# Network models reveal high-dimensional social inferences in naturalistic settings beyond latent construct models
Junsong Lu [1] ✉ & Chujun Lin [2]

Long-standing research suggests that social inferences are captured by a few latent dimensions (e.g., warmth and competence). Others argue that social inferences are more complex but lack sufficient empirical support. Here, we conducted two pre-registered studies to test the high-dimensional properties of social inferences. To maximize generalizability, we computationally sampled diverse naturalistic videos and recruited U.S. representative participants (Study 1, $N = 1598$). Participants freely described people in videos using their own words. Cross-validation identified 25 latent dimensions which explained only 15% of the variance in the data. Alternatively, a sparse network model representing the unique correlations between inferences better represented the data. The network models informed the dynamics of naturalistic inferences, revealing how different inferences co-occurred and how they unfolded over time from concrete to abstract (Study 1). The network models also indicated cultural differences in how one inference was related to another between samples (Study 2, Asian $N = 651$, European $N = 792$). Together, these findings show that the high-dimensional network approach provides an alternative model for understanding the mental representation of social inferences in naturalistic contexts, which provides new insights into the dynamics and diversities of social inferences beyond the static, universal structure found with traditional low-dimensional latent-construct approaches.

People routinely form social inferences from facial appearance and other minimal cues, such as others' personality traits, emotions and intentions, and socioeconomic status[1–4]. While the validity of these social inferences remains debated, they influence who we decide to approach and avoid, endorse as a leader, condemn as a criminal, and trust in science[5–11]. How do people organize the vast variety of social inferences in the mind?

Given the cognitive limitations of the mind, a popular belief is that social inferences can be represented by only a few latent dimensions. By analogy, all the colors humans perceive can be represented by the mixture of only three color dimensions (red, green, blue)[3,12,13]. The search for the few core dimensions of social inferences dates back to Asch's 1946 research[14], in which he found that warmth and intelligence were central because changing either one in a list of traits that described a person greatly shifted people's overall impression of the person. These core dimensions were found to have the strongest intercorrelations with other traits[15], which lays the foundation for popular statistical tools such as factor analysis for identifying the core dimensions in subsequent works.

Over the past six decades, various sets of dimensions have been proposed and empirically demonstrated[13,16–19]. For instance, Rosenberg and colleagues[20] identified three dimensions summarizing trait inferences: good–bad, hard–soft, and active–passive. Others propose only two dimensions, the Big Two models, which have various versions. For instance, Fiske and colleagues[17] found that the warmth and competence dimensions underlie stereotypes towards social groups, which depict perceived intention and capability of others, respectively. Abele and Wojciszke[12] found that the agency and communion dimensions underlie judgments of the self and others, which describe one's desire to assert the self and relate to others, respectively.

Recent work improved upon earlier literature to include more diverse stimuli and naturalistic paradigms, demonstrating an even greater number of dimensions[21,22]. For instance, using much more diverse face images and trait words that were computationally sampled, Lin and colleagues[23] found four dimensions (warmth, competence, femininity, youthfulness). By allowing participants to freely describe target individuals using their own words, Nicolas and colleagues[24] found 40 dimensions based on social group

[1]Department of Psychology, University of California San Diego, La Jolla, CA, USA. [2]Department of Psychology, Columbia University, New York, NY, USA.
✉ e-mail: jul140@ucsd.edu

labels (e.g., morality, sociability, ability, assertiveness), and Connor and colleagues[25] found 13 dimensions based on naturalistic face images (e.g., gender, sociability, age, adventurousness).

These much greater numbers of dimensions raise the question of whether social inferences in the real world, which depend on much more complex processes than those examined in past research[26], can be represented by the mixture of a relatively small number of latent dimensions (Fig. 1A). Indeed, reducing social inferences to even 40 core dimensions may not capture the complexity of social inferences. For instance, describing an individual using different words that belong to the same dimension may convey distinct impressions: a person described as affectionate and creative reflects stereotypes associated with artists while a person described as empathetic and meticulous reflects stereotypes associated with physicians, even though both affectionate and empathetic are descriptions about warmth, and both creative and meticulous about competence.

A high-dimensional mental representation, such as a network, that uniquely captures each social inference may be necessary to reflect the complex processes involved in eliciting that particular inference such as sensory inputs, stereotypes, and motivations[26,27] (Fig. 1B). Freeman and colleagues have proposed theoretical frameworks that support such a high-dimensional representation of social inferences[26,28]. However, empirical findings comparing the high-dimensional network representation and the low-dimensional latent factor representation are needed.

Nevertheless, high-dimensional network representations have gained popularity in personality research for their ability to explain individual differences in personality and psychopathology[29–34]. Instead of proposing that individual differences in behavior and cognition are driven by hidden personality dimensions (e.g., the Big Five personality traits[35,36]), the network representation explains individual differences through distinct causal links among behaviors. For example, the high correlation between "likes people" and "goes to parties often" may simply be explained by the causal link between the two—that is, if a person likes people they may then go to parties often—without assuming a hidden personality trait that causes both behaviors. While "likes people" causes some individuals to "go to parties often", it may cause other individuals to "volunteer at their community often". Thus, distinct causal links between behaviors can capture individual differences without assuming differences in hidden personality dimensions[33]. Such high-dimensional network representations may also capture mental representations of social inferences in naturalistic contexts better than the low-dimensional latent factor representations. First, in the real world, associations between social inferences may reflect associations between personalities. This is because people may rely on the social reality of personalities to inform social perception through cultural learning. For instance, exposure to different personality structures in different cultures shapes the way people conceptualize trait associations, which in turn guide social inferences[37,38]. Thus, if the network representation better captures personality, it would also capture social inferences. More broadly, this cultural learning perspective to social inferences also predicts that other factors beyond personality, such as cultural norms and language, would lead to

significant cultural variations in the network representations of social inferences[39,40].

Second, the artificial designs used in prior research may have inflated the covariance among social inferences, thereby favoring the low-dimensional latent factor representation. For instance, when participants were forced to rate the targets on a range of traits based on their faces alone, these different trait inferences may rely on very similar and limited cues from the face and thus inflating the correlations between trait inferences. However, in the real world, people observe a wealth of information streams of other people, and they use distinct cues to make different inferences[41]. Thus, the degree to which different social inferences in naturalistic contexts commonly covary—the assumption of the latent factor model—may be reduced. More broadly, different social inferences in naturalistic contexts may be elicited by different perceptual inputs, stereotypes, and motivations[26,28,41–46]. Even the same social inference may be elicited by different inputs depending on the context (e.g., "likes people" may be inferred from "goes to parties often" or "works as a caterer"). Furthermore, different social inferences may also elicit one another through heuristics (perceived attractiveness elicits perception of other positive characteristics such as intelligence[47–50]) and other conceptual biases (stereotypical relations between feminine-looking individuals and characteristics traditionally associated with females[45,51]). Thus, the complex relations between social inferences in naturalistic contexts may be better captured by high-dimensional network representations, which allow for unique variations across inferences, than the latent factor representation, which assumes a large amount of common variation across inferences.

Across two pre-registered studies, we tested the high-dimensional network model to capture mental representations of naturalistic social inferences. To elicit diverse and realistic social inferences, we computationally sampled naturalistic videos from 10,000 social media clips[52] to derive a subset of 444 videos that feature target individuals with diverse demographic backgrounds, nonverbal information, and narrative topics. Unlike static images and verbal descriptions, naturalistic videos present target individuals through multi-modal information streams dynamically, which closely resemble the way people observe each other in real life. Participants freely described these target individuals using their own words at three different time points of the videos. Compared to traditional paradigms such as forced ratings on a fixed list of traits selected by researchers, unconstrained descriptions allow for capturing social inferences that spontaneously occur and naturally shift over time.

In Study 1, we tested the high-dimensional network in comparison to the low-dimensional latent representation using social inferences made by a large, representative U.S. sample ($N = 1598$). We first analyzed the performance of the latent representation. We then assessed the performance of the network representation relative to the latent representation. After establishing the validity of the network representation, we showed how the interconnections between social inferences in the network offer an alternative understanding of the core "dimensions" and perception dynamics. To test the generalizability of our findings, in Study 2, we recruited

**Fig. 1 | Two models describing mental representations of social inferences. A** The low-dimensional latent construct model. Orange dots indicate social inferences. Gray dots indicate latent dimensions. Straight lines indicate the associations between social inferences and latent dimensions. Broken curves indicate the associations between latent dimensions. **B** The high-dimensional network model. Orange dots indicate social inferences, and lines (e.g., the highlighted blue line) indicate unique associations between them. Note that both models describe the underlying psychological structure of social inferences instead of personality traits, since perceivers' inferences may not reflect targets' true characteristics.

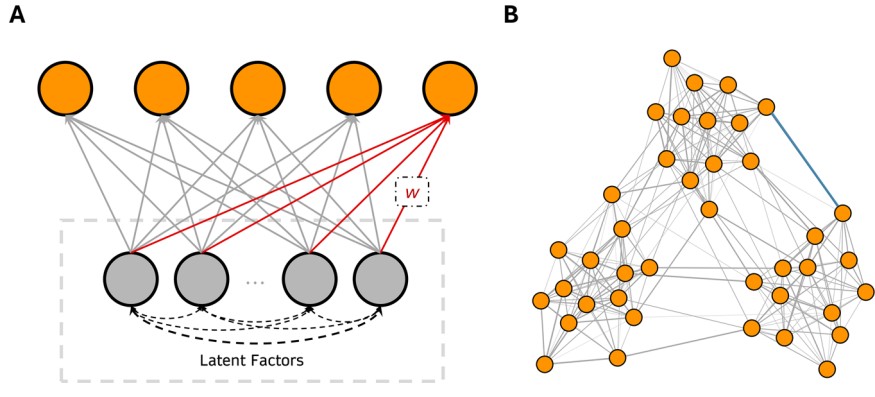

additional samples from two distinct world regions, Asia ($N = 651$) and Europe ($N = 792$), and compared the performance of the network and the latent representations in each sample.

## Methods

All studies were pre-registered via the Open Science Framework. Study 1 was pre-registered on February 8, 2024 (https://osf.io/43f8x/?view_only= c6f4eac58ea940f09e444c6aed27830e), and Study 2 was pre-registered on August 7, 2024 (https://osf.io/s5vpn/?view_only=7f7dd0b463554c12976 6a3a3d41cc9a5). We followed all pre-registered methods, conducted and reported all results from pre-registered analyses. In addition, we conducted two sets of exploratory analyses in Study 1, including the direct comparison between the two models (quantification as well as visualization) and a further analysis of the properties of the network (model fit, community detections, and network dynamics). We did not conduct any exploratory analyses in Study 2.

### Ethics

Both studies were approved by the Institutional Review Board at the University of California, San Diego (808857), and all participants provided informed consent in accordance with approved procedures.

### Study 1

**Stimuli**. We aimed to create a video set that represented diverse social information streams similar to those people encounter in daily life. We first determined the number of videos that we needed. We expected to partially replicate some of the dimensions found in prior research. To detect a small correlation ($r = 0.17$)[53] between the factor scores of two dimensions with 95% power and a significance level of 0.05, a correlation power analysis test indicated that we needed 444 videos. To select these 444 videos from as large and diverse set of videos as possible, we targeted the First Impression video dataset that were gathered in prior research[52] (https://chalearnlap.cvc.uab.cat/dataset/24/description/). This dataset contains 10,000 clips from 3000 + HD YouTube videos. The videos show people of diverse ages, genders, and races speaking freely in front of the camera on various topics (e.g., personal life, entertainment, education) in a range of environments (e.g., indoors, outdoors). These videos varied in length. To minimize our experiment time while ensuring that we provided sufficient information for people to make various social inferences, we trimmed all videos to the first 10 s.

To make sure that the 444 videos we selected from these 10,000 videos preserved the diversity of the original dataset, we followed a two-stage selection procedure (Fig. 2). First, we manually excluded videos based on 9 criteria to ensure that the remaining videos were of high audiovisual quality and portray one target person clearly. We excluded videos that (i) were of low visual resolution; (ii) were of low audio quality; (ii) were in black and white; (iv) were distorted vertically or horizontally; (v) show target persons under the age of 18; (vi) showed very little body parts of the target person; (vii) showed more than one target person talking; (viii) showed text boxes that blocked the view of the target person; and (ix) were shorter than 10 s.

The above exclusion criteria retained 1897 videos. To further select a diverse subset of 444 videos, we applied computational tools to sample videos of distinct target individuals that conveyed maximally different demographic, visual, auditory, and narrative information. We first quantified multi-modal features of the targets in each video, including the visual features of facial structure (using the InsightFace[54] algorithm), facial action units, inferred emotions, head pose (Py-Feat[55]), the auditory features of frequency, spectrum, and amplitude (OpenSmile[56]), and the narrative contents quantified as word embeddings (deBERTa[57]), with a total of 2166 features. We then grouped the 1897 videos based on age (younger than 33 years old, older than 33 years old), gender (female, male), and race (Asian, Black, and White), resulting in 12 groups. The demographic information was provided by the First Impression video dataset, annotated using pretrained deep learning models with relatively low errors. However, there may still be a slight mismatch between these computational annotations and the

targets' own self-identified age, gender, and race. The Asian category collapsed across East Asian, South Asian, and Southeast Asian targets. Finally, for each age-by-gender-by-race group, we sampled videos that maximally differed in their visual, auditory, and narrative information using the maximum variation sampling procedure[23]. The number of videos sampled per group was proportional to the total number of videos of that group in the 1897 videos. To address the issue that higher-dimensional features (e.g., word embedding with 1536 features) received greater weights in the selection process than lower-dimensional features (e.g., head pose with 3 features of roll, pitch, yaw), we L-2 normalized the features before sampling.

We confirmed that the resulting 444 videos from the above selection procedure were diverse in demographic information (Table 1) and multimodal information (Fig. 2B histogram in the lower-right corner). To validate the representativeness of the 444 videos in terms of multi-modal information streams, for each video, we calculated the variance for each of the 2166 features and the mean of these variances across features. We compared this mean variance in the multi-modal information for the selected 444 videos to a random set of 444 videos (randomly picked 1000 times) using a one-sample $t$ test, which showed that our selected 444 videos had larger variance across multi-modal information streams than videos that were randomly picked, $t(999) = 396.25$, $P < 0.001$, Cohen's $d = 12.53$, 95% CI [11.97, 13.08].

**Participants**. As pre-registered, we recruited participants who were representative of the United States population along the gender, age, and race distributions. We included participants who were: (i) located in the United States; (ii) aged 18 and above; (iii) with at least a high school education; (iv) with normal or corrected-to-normal vision and hearing; and (v) native English speakers.

We determined our sample size based on formal power analysis using data from prior research[23]. In that study, 30 participants freely wrote descriptions about 100 faces. Since the present study would be using aggregated data, we estimated how many participants we needed to obtain a satisfactory consensus among participants using the prior data. To calculate the between-subject consensus in the descriptions participants wrote, we constructed a word frequency matrix with participants in the rows and words in the columns. The consensus between two participants was assessed with the Spearman correlation between their word frequency (the columns in the word frequency matrix). The average pairwise consensus between participants was 0.196 based on the prior data, which was converted to a standardized Cronbach's alpha of 0.879. Targeting an alpha of 0.90, the Spearman-Brown forecasting formula indicated that we needed 37 participants to evaluate each video. Assuming a 15% participant-wise exclusion rate, we decided to recruit at least 44 participants for each video.

Since it would take too long for each participant to evaluate each of the 444 videos, we randomly assigned these videos into 37 modules, each with 12 videos. Thus, it required 1628 participants across 37 modules in total, with each module evaluated by 44 participants. We recruited participants via CloudResearch Connect. A total of 1640 participants were recruited (Table 2). We processed the data based on pre-registered exclusion criteria: (i) we excluded a word if it was meaningless; and (ii) we excluded participants who did not meet our inclusion criteria, failed one or more attention checks (see Procedures below), or had more than half of their videos excluded due to providing only meaningless words as responses. Based on these exclusion criteria, we were left with a final set of 1598 participants (53.3% female, age ($M = 41.9$, SD = 13.3), 77.3% White, 13.60% Black, 7.01% Asian, and 2.07% others; see Table 2).

**Procedures**. Each participant was randomly assigned to complete one module of the experiment, which took ~26 min. To alleviate the effect of memory load, we allowed participants to write down their spontaneous inferences of the target individual in each video across three different time points: one at the beginning based on the first frame, the second time in the middle of the video (5 s), and finally at the end of the video. At each of the three pauses, the video stopped playing and participants were asked

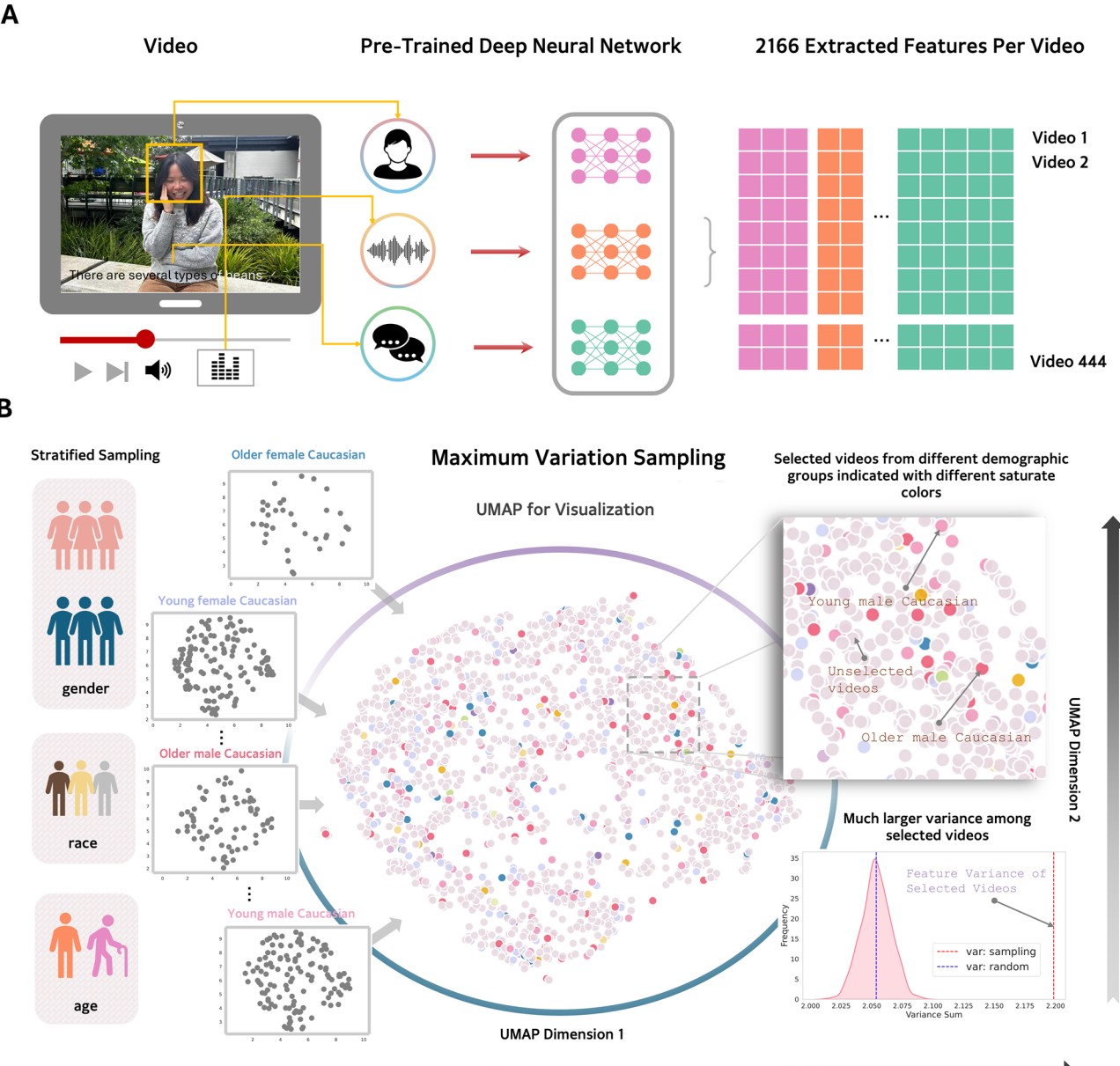

**Fig. 2 | Selecting videos of diverse target individuals and multi-modal information. A** Quantifying multi-modal social information (visual, purple; auditory, orange; speech, green) for each video using pre-trained deep neural networks. **B** Uniform manifold approximation and projection (UMAP) of the 444 selected videos (color dots) and the rest of the videos (gray dots) based on the multi-modal information quantified in (**A**). We applied maximum variation sampling to select 444 videos that were maximally different in multi-modal information for each age-

by-gender-by-race group (zoom-in plots). Compared to random sets of 444 videos (*n* = 1000 samples), the selected videos showed significantly greater variance across multi-modal information streams. **A** Includes a representative photograph of an identifiable individual, used for illustrative purposes only and not part of the actual stimuli. The person depicted has provided written consent for publication. Photo credit: Chujun Lin.

**Table 1 | Summary of demographic information of target individuals in videos**

| Video set | #Videos | #Targets | Power analysis | Gender | Age | Race |
|---|---|---|---|---|---|---|
| Videos surviving the exclusion criteria | 1897 | 655 | N/A | 337 women 318 men | 19–24 years: 155 25–32 years: 330 33–45 years: 145 46–60 years: 16 61+ years: 9 | 562 White 75 Black 18 Asian |
| Videos selected based on multi-modal information | 444 | 444 | *N* = 444 to detect a medium correlation of 0.17 | 215 women 229 men | 19–24 years: 96 25–32 years: 223 33–45 years: 109 46–60 years: 10 61+ years: 6 | 383 White 52 Black 9 Asian |

**Table 2 | Summary of participant demographics**

| Study | Power analysis | Pre-registered plan | Excluded | Total | Final | Gender | Age | Race |
|---|---|---|---|---|---|---|---|---|
| Study 1 | $N \geq 37$ for each of the 37 modules | 44 per module if exclusion <15% | 2.56% participants (n = 42); 0.057% pauses (n = 33); 0.25% descriptions (n = 562) | 1640 | 1598 participants; 57,495 pauses; 225,184 descriptions | 851 (53.3%) women; 743 (46.5%) men; 4 (0.25%) non-binary | Min = 18; Max = 86 M = 41.9 SD = 13.3; 327 (20.5%) Aged 18–29; 626 (39.2%) Aged 30–44; 449 (28.1%) Aged 45–59; 196 (12.3%) Aged 60+ | 1235 (77.3%) White; 218 (13.6%) Black; 112 (7.01%) Asian; 33 (2.07%) Other |
| Study 2 (Asia) | $N \geq 37$ for each of the 15 modules | 44 per module if exclusion <15% | 3.84% participants (n = 26); 0.081% pauses (n = 19); 0.55% descriptions (n = 534) | 677 | 651 participants; 23,413 pauses; 97,115 descriptions | 315 (48.4%) women; 332 (51.0%) men; 4 (0.61%) non-binary | Min = 18; Max = 71 M = 30.0 SD = 8.55; 373 (57.3%) Aged 18–29; 232 (35.6%) Aged 30–44; 42 (6.5%) Aged 45–59; 4 (0.6%) Aged 60+ | 106 (16.3%) White; 1 (0.02%) Black; 543 (83.4%) Asian; 1 (0.02%) Other |
| Study 2 (Europe) | $N \geq 37$ for each of the 15 modules | 44 per module if exclude <15% | 1.73% participants (n = 14); 0.049% pauses (n = 14); 0.27% descriptions (n = 280) | 806 | 792 participants; 28,498 pauses; 103,966 descriptions | 389 (49.1%) women; 398 (50.3%) men; 5 (0.63%) non-binary | Min = 18; Max = 82 M = 40.3 SD = 13.6; 181 (22.9%) Aged 18–29; 341 (43.1%) Aged 30–44; 187 (23.6%) Aged 45–59; 83 (10.5%) Aged 60+ | 676 (85.4%) White; 60 (7.6%) Black; 56 (7.10%) Asian |

to freely write down any word that they thought of the person based on the information accumulated so far. For each pause, participants were shown a list of ten input boxes and were instructed to write one word in each box. They only needed to write at least one word at each pause to proceed. After participants confirmed the entries, the video resumed to play or moved on to the next video if they already completed all three pauses for the current video. Two attention checks were randomly interspersed between the third and twelfth trials. In each attention check, participants viewed a 2-second video clip, either from the last three trials or a new video and indicated whether they had seen the video earlier in the experiment.

**Validation of designs**. To validate that participants' free descriptions of our selected videos covered a wide range of social inferences, we examined the topics revealed in these descriptions using topic modeling. As pre-registered, we first used one of the most popular approaches for topic modeling, BERTopic[58]; in addition, we also validated our data using a newer state-of-the-art method, TopicGPT[59]. In brief, topic modeling takes in inputs of documents (e.g., the free descriptions mentioned for a video), quantifies these documents using word embeddings (i.e., numbers that represent the meaning of the documents), and groups these documents into different topics based on their word embeddings' similarities (e.g., using methods such as clustering).

Before conducting topic modeling, we first preprocessed the descriptions to remove meaningless words (e.g., words participants typed by randomly pressing different keys), correct for spelling errors, and lemmatize each word to their base form (e.g., converting both "creativity" and "creative" to "creative"). To reduce noise, we excluded descriptions that were reported by only one participant or in less than 1% of the videos (i.e., 4 videos). We then obtained the word embeddings of the preprocessed descriptions using the state-of-the-art algorithm by OpenAI (text-embedding-3-small). For clustering these documents into topics, we optimized the hyperparameters of BERTopic using Bayesian Optimization[60]. This analysis indicated that participants' free descriptions of the targets in the videos indeed covered a wide range of 16 topics (Fig. S1A). However, this method failed to identify the topics for many descriptions (gray dots in Fig. S1A). Thus, we also validated our data using the state-of-the-art TopicGPT method[59]. This method successfully categorized most of the descriptions into topics and identified 52 meaningfully interpretable topics (Fig. S1B). Together, these analyses confirmed that our study paradigms using naturalistic videos and free descriptions elicited a rich set of social inferences.

**Exploratory factor analysis (EFA)**. As pre-registered, we used the exploratory factor analysis to investigate the latent constructs underlying social inferences. EFA is one of the most popular methods in the literature for uncovering latent constructs. We first converted the preprocessed descriptions into a document-term matrix, where the rows represented the videos and the columns represented the words participants mentioned. Each value in this matrix indicated how frequently a word in a particular column was mentioned by participants who watched the video in a particular row across all three pauses. To account for the imbalanced numbers of participants who viewed different videos, we normalized the frequency using min-max normalization. To adjust the data distribution towards a normal approximation, we applied a nonparanormal transformation. We then computed the Spearman correlations between words based on their occurrence frequency. Words that exhibited similar distributions of frequencies across videos had greater correlations. To determine the optimal number of latent dimensions underlying our data, we used bi-cross validation (BCV) for 100 iterations. BCV outperformed traditional methods for determining the number of underlying dimensions, such as Horn's parallel analysis[61]. Given the optimal number of dimensions identified, we then conducted EFA to extract to dimensions and examine their interpretations.

**Sparse network modeling (SNM)**. We used the sparse network model to investigate the high-dimensional representations underlying social inferences. SNM also captures the efficiency of the mind by only modeling the associations between social inferences that are important and cannot be explained by the indirect associations through other social inferences[62]. Specifically, in our case, each node in the SNM represented a social inference made by our participants, and each edge between two nodes represented the partial correlation between the two social inferences after partialling out their connection through all the remaining social inferences. To encourage efficient representation, SNM used graphical LASSO to penalize unnecessary connections between social inferences[62,63]. The regularization hyperparameter $\gamma$ used to control the preference over simpler models was specified as 0.25 for balancing discovery and parsimony of meaningful connections.

**Comparing the exploratory factor model and the sparse network models**. To compare how well the latent construct representation and the high-dimensional network representation explain our data, we used the resulting EFA model and the SNM model to reconstruct the correlation matrix between social inferences. In brief, to reconstruct the correlation matrix between social inferences, for the EFA model, we multiplied the factor loading matrix by the factor correlation matrix and then by the transpose of the factor loading matrix; for the SNM model, we inverted the precision matrix which indicated conditional independence among social inferences. We compared the reconstructed correlation matrix from each of the two models to the actual observed correlation matrix between social inferences in our data. We quantified the reconstruction error for each model using the Standardized Root Mean Square Residual (SRMR). To estimate the robustness of the SRMR, we followed Van Bork (2021)[64] to resample the correlations 100 times from the reproduced covariance matrix, obtaining a sampling distribution of SRMR.

**Investigating the properties of the high-dimensional network representation**. In addition to comparing the performance of the SNM model relative to the EFA model, we also assessed the absolute fit of the SNM model to our data. To this end, we used confirmatory network analysis within a structural equation modeling framework[65]. This analysis first estimated an SNM, based on which we reconstructed the correlation matrix as described above; then we used this reconstructed correlation matrix to specify the structural equation model between social inferences; finally, we fitted the structural equation model to the observed correlation matrix. This procedure produced standard model fit indices in the structural equation modeling, such as the Comparative Fit Index (CFI), Tucker-Lewis Index (TLI), and Root Mean Square Error of Approximation (RMSEA), with CFI and TLI values closer to 1 and RMSEA values below 0.06 indicating a good fit of the SNM to the data[66]. The computation of this procedure is highly demanding, we thus conducted this analysis for a subset of the top 100 social inferences that were mentioned in most of the videos, which already took weeks to run.

After confirming the model fit of the network representation, we further investigated the properties of social inferences revealed by this representation. First, we examined the communities detected in the correlation matrix between social inferences estimated based on the SNM. A community in a network is a group of nodes (i.e., social inferences) that are highly connected with each other and minimally connected with the rest of the nodes outside of the group[67]. Since the SNM was estimated based on the co-occurrence between inferences across videos, the communities informed the inferences that tended to co-occur in a video. Although both communities and latent constructs seem to describe core groups of social inferences, they are mathematically distinct. Communities describe what social inferences are highly correlated (e.g., which words people often use to describe the same individual); whereas latent constructs describe why certain social inferences are highly correlated (e.g., gender stereotype). We performed community detection using the Louvain modularity algorithm[68].

Beyond investigating the occurrence of words across videos, we also examined the occurrence of words over time. In particular, we investigated how the connections between inferences changed as participants wrote down their spontaneous inferences across different time points of the videos. We first grouped the inferences into four time points: the first words that participants wrote at the first pause (based on the first still frame of the video), all the rest of the words participants wrote in the first pause, all the words participants wrote in the second pause (after watching half of the video), and all the words participants wrote in the third pause (after watching the entire video). Then, for social inferences within each time point, we estimated an SNM model. For each model, we calculated the strength centrality of each social inference (i.e., a node's importance as indicated by the sum of the weights of all edges connecting this node with other nodes in the network) based on the reconstructed correlation matrix. Finally, we computed the change of centrality for each social inference between the current time point and the previous time point, which indicates the dynamic emergence of the most important social inferences that participants wrote down at the current time point.

## Study 2

After demonstrating the validity of the high-dimensional network model for describing the mental representation of social inferences in naturalistic contexts in Study 1, we next tested the generalizability of this model to capture social inferences made by participants in different world regions.

**Stimuli**. Based on Study 1, we found that a smaller subset of the 444 videos was sufficient to capture the observed covariance structure of social inferences. Specifically, to estimate the optimal number of videos that we needed to sufficiently capture the relations among social inferences found with a large set of videos, we gradually removed the videos randomly from our dataset and recalculated the covariance matrix among social inferences based on the remaining videos. We then computed the correlation between the covariance matrix based on the remaining videos and the covariance matrix based on the full set of 444 videos. This analysis indicated that with 175 videos, we would be able to reproduce the covariance matrix based on the full video dataset with a strong correlation (Spearman's rho >0.6). Considering an even number of videos in each module (each module showing 12 videos as in Study 1), we included 180 videos from the 444 videos in the current study. These 180 videos were selected based on the same maximum variation sampling procedure as in Study 1 to ensure the diversity of the targets' demographic characteristics and the social information they conveyed. These 180 videos were randomly assigned to 15 modules, each with 12 videos.

**Participants**. As pre-registered, we recruited participants from both Asian and European countries via Prolific. We included all countries within each region that were available on Prolific, covering 48 countries/administrative regions in Asia and 44 in Europe. We applied the same inclusion criteria as in Study 1, except that participants were required to locate in the corresponding regions instead of the U.S. and could be either native English speakers or proficient in English. We followed the same sample size, 44 participants per video (per module), as in Study 1.

In total, we recruited 677 participants in Asia and 806 participants in Europe (extra participants were recruited in Europe due to randomization hiccups). We applied the same exclusion criteria as in Study 1. After exclusion, we were left with a final set of 651 Asian participants (48.4% female, age ($M = 30.0$, SD = 8.55), 16.3% White, 0.02% Black, 83.4% Asian, and 0.02% others) and 792 European participants (49.1% female, age ($M = 40.3$, SD = 13.6), 85.4% White, 7.60% Black, and 7.10% Asian; see details in Table 2).

**Procedures**. The procedures were identical to Study 1.

**Analysis methods**. We repeated the analyses of EFA (Study 1, Analysis I), SNM (Study 1, Analysis II), and model comparison between them

(Study 1, Analysis III) as in Study 1 using data from each sample in Study 2.

Using the datasets from Study 1 (U.S. sample) and Study 2 (samples from Asia and Europe), we investigated differences in the properties of social inferences revealed by the network representation. We compared each of the samples in Study 2 with the U.S. sample in Study 1 using the network comparison test[69]. A significant difference in an edge between two nodes indicates that the representation of how the two social inferences are uniquely linked is different between the participants in these two samples[70]. As pre-registered, for each pair of the targeted samples, we identified the top 300 descriptions that were most frequently mentioned across the videos and that overlapping between the two samples. Based on these shared descriptions, we estimated the SNM for each sample. To conduct the network comparison test, we merged the two document-term matrices from the two samples and resampled the word frequencies per document from this merged matrix 1000 times. In each resampling iteration, we resampled two sets of entries to fill two null document-term matrices that matched the size of the original document-term matrices from each sample; for each null document-term matrix, we sampled with replacement the rows from the merged document-term matrix; after filling each of the null document-term matrices, we refitted an SNM for each null document-term matrix; then given these two null SNMs, we identified the largest absolute difference in the edge weights between the two SNMs among all the edges. Across 1000 iterations, we obtained a null distribution of the maximal differences in the edge weights between the two SNMs. Finally, we assessed the statistical significance of the observed difference of each edge between the two samples by computing how many of the null maximal differences were greater than the observed difference. This method compares every edge between two networks while correcting for multiple comparisons across all the edges.

## Results
### Study 1
After data exclusion (i.e., removing meaningless words and participants who failed attention checks), we obtained a total of 225,184 descriptions across participants, which consisted of 9362 unique words. On average, each participant wrote 2.91 words (SD = 2.34) in each pause (8.74 words per video across the three pauses). In the preprocessing, we excluded words that only one participant mentioned and mentioned in less than 1% of the videos, obtaining 2964 unique words. The 1% exclusion criterion was based on common thresholds in the literature of natural language processing[71]. After lemmatization to convert words to their base form, we obtained 2926 words, which we used to conduct the remaining analyses. Unlike prior studies[72], we did not observe a significant social desirability bias, suggesting more diverse trait inferences in naturalistic settings. Sentiment analysis, which assigns each word a score ranging from −1 (negative) to 1 (positive), revealed a mean valence close to zero (US mean = 0.08), with a wide spread of valence score and most words being neutral.

Words mentioned per video were aggregated across participants to create the document-term matrix (i.e., video–word matrix). We computed the Spearman's rank correlations between words based on this document-term matrix, which ranged from −0.84 to 0.96 across word pairs.

**A common set of latent constructs insufficient for capturing naturalistic inferences.** Following the popular belief that a small number of latent constructs underlie social inferences, we first examined these latent constructs. Bi-cross validation with 100 iterations indicated that 25 dimensions optimally described our data (Fig. 3). We then performed exploratory factor analysis (EFA) to extract 25 dimensions to examine their content. We found that these dimensions were interpretable and overlapped with those proposed by prior theories[25,41,50] (e.g., gender perception, physical fitness, positive affect, leadership, introversion; dimension labels generated using GPT-4o based on the top five loaded words; see Fig. S2 for all dimensions). These 25 dimensions were weakly correlated, with correlation coefficients ranging from −0.16 to 0.17. These findings suggest that a much larger number of latent constructs

were needed to account for social inferences formed in more naturalistic contexts.

However, the variance in the data captured by these 25 dimensions was low. The proportion of common variance explained by each dimension ranged from 3 to 6%. In total, only 15% of the total common variance in the data was explained. To verify whether this low amount of common variance explained was due to the smaller number of dimensions compared to the large number of inferences in our data, we tested a 100-dimension solution. However, even with 100 dimensions, the model only captured 39% of the common variance in the data. These results suggest that the complex relations between social inferences formed in naturalistic contexts may not be sufficiently explained by only a common set of latent constructs.

**Network representation captures complex relations between naturalistic inferences.** The above results raise the possibility that the relationships between social inferences in naturalistic contexts may be underlain by mechanisms unique to the specific pairs of social inferences involved and are unreducible to the same set of latent constructs. To test this possibility, we modeled our data using a sparse network model (SNM), which captured the unique relationship (i.e., partial correlation) between pairwise social inferences that were unexplainable through other social inferences. We compared the performance of SNM to that of EFA by assessing how well they each recovered the correlations we observed in the data between social inferences. We found that the SNM model had a much smaller error (standardized root mean square residual, SRMR = 0.007) than the EFA model (SRMR = 0.043) across the 100 resampling of the correlations from the reproduced covariance matrices, $t$ (125.71) = 1285.50, $P < 0.001$, Cohen's $d = 181.80$, 95% CI [163.98, 199.62]. We visualized this quantitative comparison by plotting the observed correlations and the recovered correlations based on each model using networks (Fig. 4; the layout of the networks was determined using the ForceAtlas2 algorithm[73]). The visualization indicated that the social inferences in our observed data were densely connected, which was more closely captured by the SNM; whereas, the latent constructs in the EFA only captured the connections between a subset of the social inferences, leaving the connections with the rest of the social inferences largely unexplained.

We also directly assessed the performance of the SNM in capturing the relations between social inferences in our data without comparing it to the latent construct model (EFA). A confirmatory network analysis (using the top 100 social inferences, due to the extremely high computational demands of this analysis) indicated that the SNM fitted our data well: root mean square error of approximation (RMSEA) = 0.030, 90% CI [0.028, 0.032], Tucker-Lewis Index (TLI) = 0.87, comparative fit index (CFI) = 0.89. Together, these results indicate that the high-dimensional network representation offers a valid alternative beyond the low-dimensional latent constructs for modeling the relations between social inferences in naturalistic contexts.

**Network representation reveals the dynamics of naturalistic inferences.** Given the promises of the network model for capturing the mental representation of naturalistic social inferences, we further analyzed the network for insights into these inferences. We first analyzed the densely connected regions, i.e., the communities, based on the SNM using the Louvain Modularity algorithm. We found seven communities in the recovered correlation matrix between social inferences based on the SNM (Fig. 5). The number of social inferences included in each of the communities ranged from 185 to 630. These communities revealed inferences that people tended to make together for a video. For instance, if participants wrote that a person in a video appeared young, they were more likely to also describe them as knowledgeable and leader-like (Fig. 5, pink). Similarly, if participants inferred that a person in a video looked shy, they were more likely to also describe them as depressed and lonely (Fig. 5, green). These findings suggest that inferences traditionally thought to belong to different latent constructs (e.g., "young" in the

**Fig. 3 | Factor loadings of exploratory factor analysis in Study 1.** Exploratory factor analysis on the co-occurrence of the 2926 social inferences (rows, only the top five inferences per dimension are plotted here) freely generated by participants based on 444 naturalistic videos indicated 25 dimensions (columns) optimally underlie the data ($n = 444$ videos). Positive loadings are annotated in red; negative loadings in blue. The darker the color is, the greater the absolute value of the factor loading is.

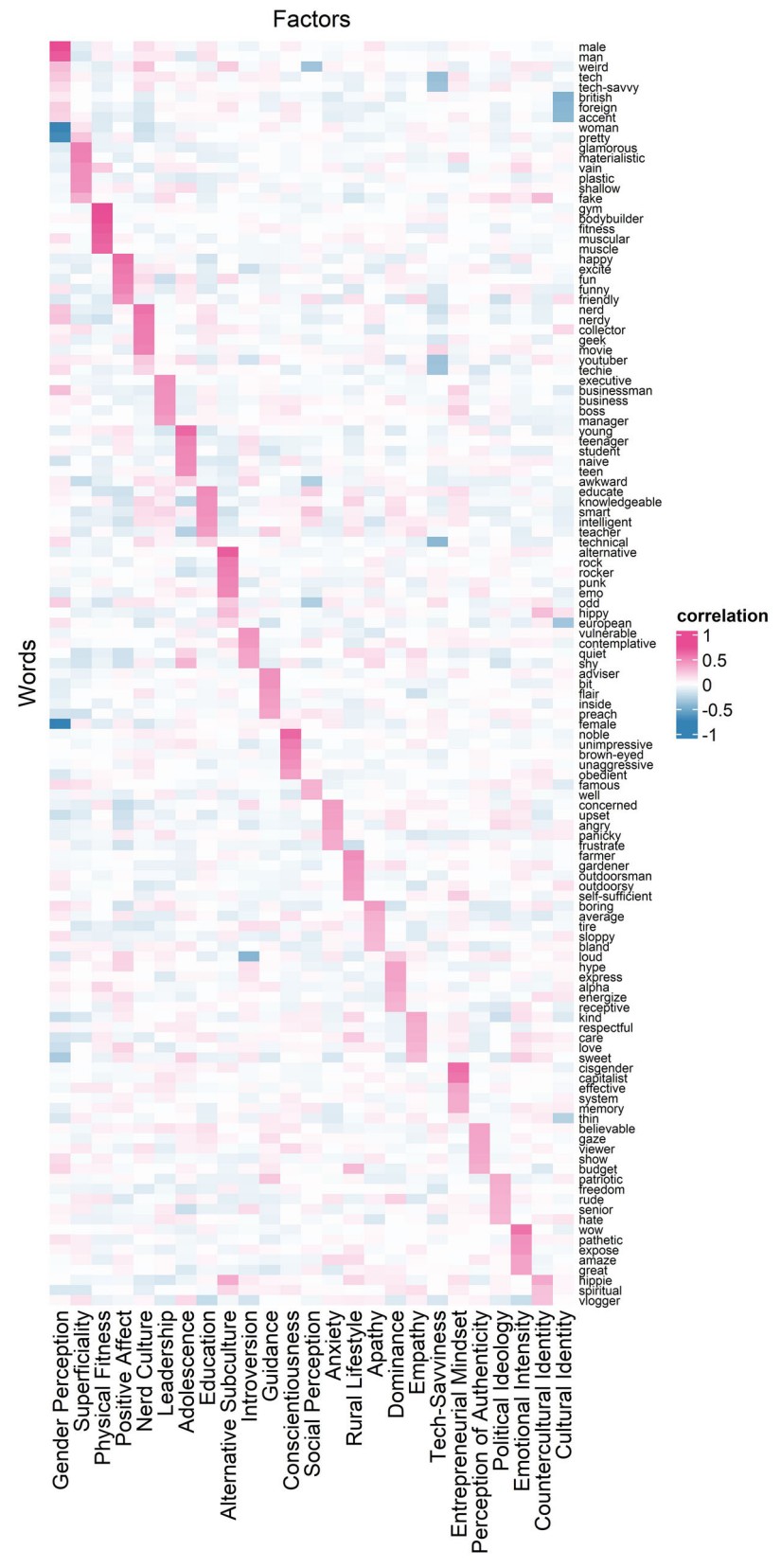

youthfulness dimension, "leader" in the competence dimension) may elicit one another in naturalistic contexts.

The network representation not only revealed the dynamics of social perception in terms of what descriptions people wrote together regarding a target but also what descriptions people wrote over time. We estimated an

SNM for inferences entered at four different time points: the first word participants wrote based on the first frame of the video, additional words they wrote based on the first frame, the words they wrote after viewing half of the video, and the words they wrote after viewing the entire video. For each network, we computed the strength centrality of each inference, which

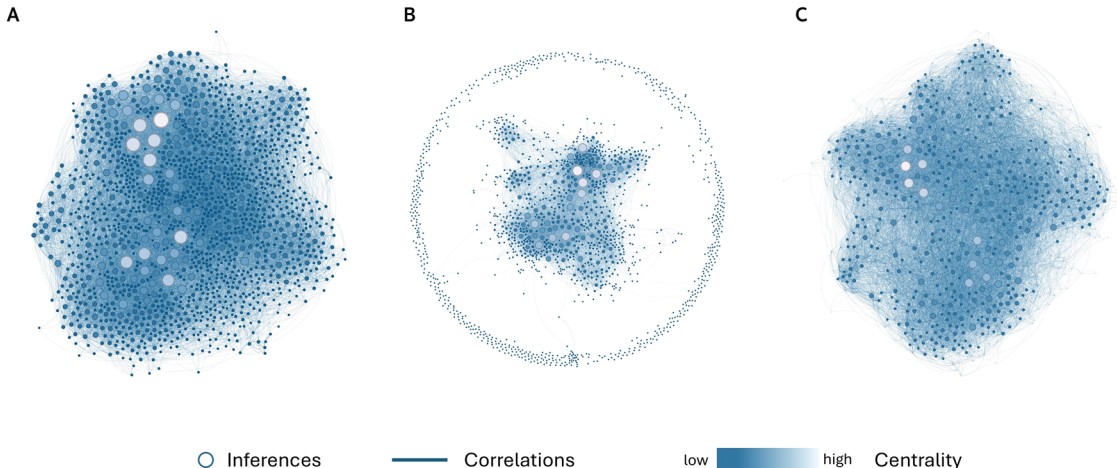

| ○ Inferences | —— Correlations | low ▨ high Centrality |

**Fig. 4 | Visualization of observed and predicted correlations between inferences.**
**A** Observed Spearman correlations between social inferences represented as a network. Each node represents a social inference spontaneously written down by our participants. The color and size of the nodes indicate the node's weighted degree (i.e., the sum of its correlations with the rest of the nodes): lighter color and bigger size indicate greater weighted degree. The edge between two nodes represents the Spearman correlation between the two social inferences computed based on their co-occurrence across the videos ($n = 444$ videos). **B** Predicted correlations between

social inferences recovered based on the 25-dimensional solution from the exploratory factor analysis ($n = 444$ videos). **C** Predicted correlations between social inferences recovered based on the sparse network model. All network graphs were plotted based on the same set of parameters (e.g., graph size area). For illustration purposes, in these network visualizations, we plotted the top 1500 words that were most frequently mentioned across participants and videos ($n = 444$ videos), and the correlations greater than or equal to 0.15.

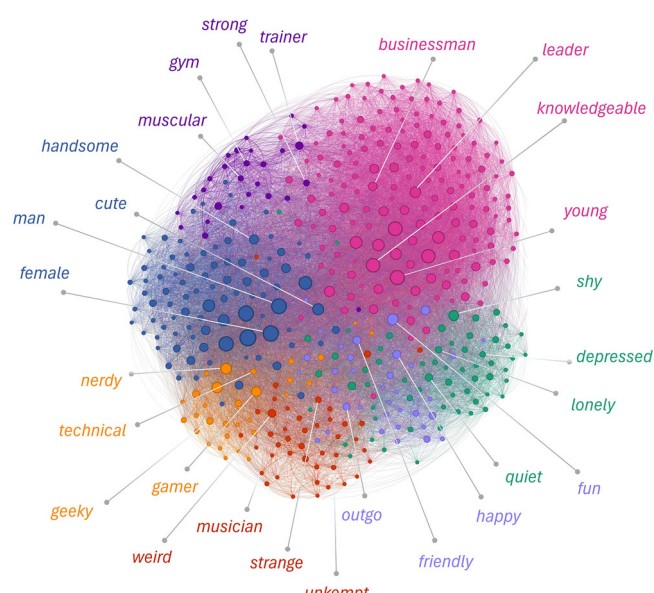

**Fig. 5 | Communities from predicted correlations of the sparse network model.**
We plotted the correlations between the top 1500 inferences most frequently mentioned across videos ($n = 444$ videos) as predicted by the sparse network model. The sparse network model was fitted based on the co-occurrence of the inferences across videos. Each node indicates a social inference spontaneously written by our participants. The size of the nodes indicates weighted degree (i.e., the sum of a node's correlations with the rest of the nodes). The edge between nodes indicates the correlation between two social inferences. For illustration purposes, only edges with weights greater than 0.10 were shown. The colors indicate the different communities identified using the Louvain algorithm. For each community, a subset of the nodes was labeled for interpretability.

is the sum of all edges connecting that inference to other inferences. Using these strength centralities, we assessed the change in connections between social inference across the four time points (Fig. 6). We found a greater increase in strength centrality of physical appearances/fitness (e.g., muscular) and gender identity (e.g., female) from time point 1 to time point 2, of

speech content/rhetorical skill (e.g., mentor) from time point 2 to time point 3, and of abstract personalities/power (e.g., executive) from time point 3 to time point 4. These findings suggest that the social inferences that were central to generating other new inferences changed over time, shifting from more physical inferences to more abstract inferences. Moreover, physical inferences continued to show high strength centrality across all four time points, suggesting their sustained influence on the descriptions perceivers wrote across all time points.

## Study 2
**Network representation captures naturalistic inferences across world regions.** After establishing the validity and utility of the network representation for understanding naturalistic social inferences in Study 1, we next tested the generalizability of this representation in other world regions. After data exclusion, data preprocessing, and lemmatization, we obtained 1351 unique words from the Asian sample and 1319 unique words from the European sample. Consistent with Study 1, sentiment analysis revealed no evidence of social desirability bias, with mean valence scores close to zero in both samples (Asian mean = 0.08; European mean = 0.09). Bi-cross validation indicated that 9 latent dimensions optimally explained the Asian data and 13 latent dimensions optimally explained the European data. As in Study 1, all these dimensions were meaningfully interpretable (e.g., gender and appearance, social anxiety, professionalism, physical fitness for the Asian sample; gender stereotypes, introversion, positive affect, intellectual competence for the European sample; labels generated by GPT-4o; see all dimensions in Figs. S2 and S3). However, again, these latent constructs only explained a small amount of common variance in our data (15% for the Asian sample; 26% for the European sample).

Corroborating findings in Study 1, we found that the sparse network model better captured the correlations between social inferences in both samples than the latent construct models (Fig. 7). For the Asian sample, the prediction error of SNM (SRMR = 0.008) was significantly lower than that of EFA (SRMR = 0.07) across the 100 resampling of the correlations from the reproduced covariance matrices, $t(118.13) = 457.94$, $P < 0.001$, Cohen's $d = 67.76$, 95% CI [58.41, 71.12]. For the European sample, the prediction error of SNM (SRMR = 0.01) was also significantly lower than that of the EFA (SRMR = 0.06): $t(163.51) = 215.67$, $P < 0.001$, Cohen's $d = 30.50$, 95%

**Fig. 6 | Changes in strength centrality per social inference over time.** We estimated four separate network models based on inferences made at four time points, resulting in networks with different numbers of social inferences ($n$ = 523, 1588, 2339, and 2926 social inferences). All networks were constructed from the same set of videos ($n$ = 444 videos). The rows showed a subset of social inferences with the largest increases in strength centrality across time points (top five per time point, with a total of 19 unique social inferences across the four time points). The color of the dots indicates the increment in strength centrality relative to the previous time point. The size of the dots indicates the strength of centrality at the current time point. Inferences in the rows are sorted from bottom to top in descending order according to the increase in centrality at each time point.

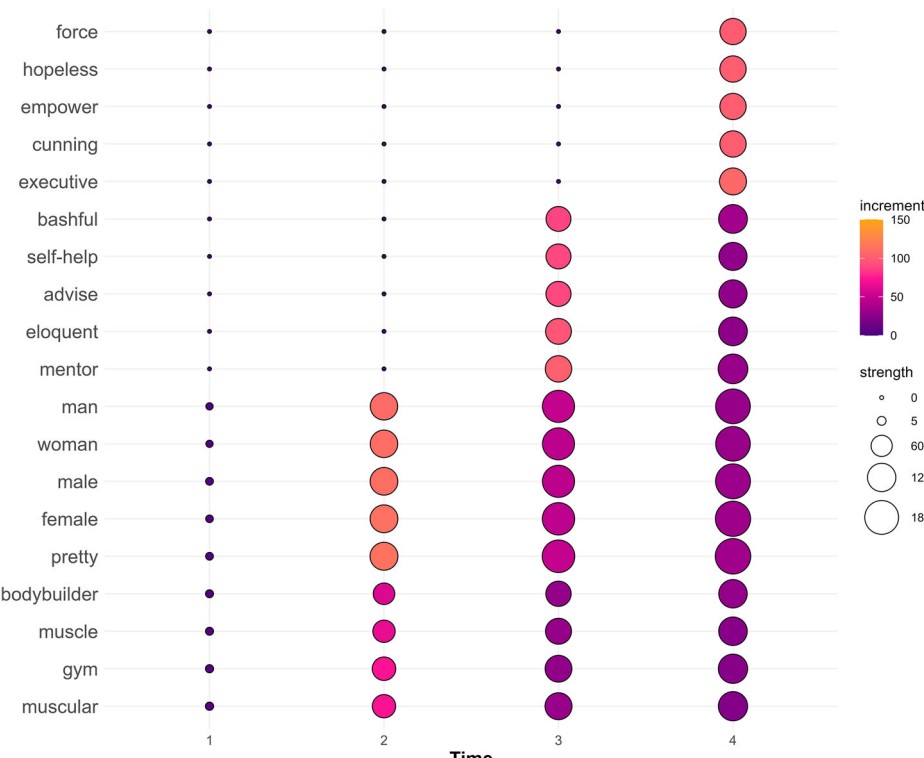

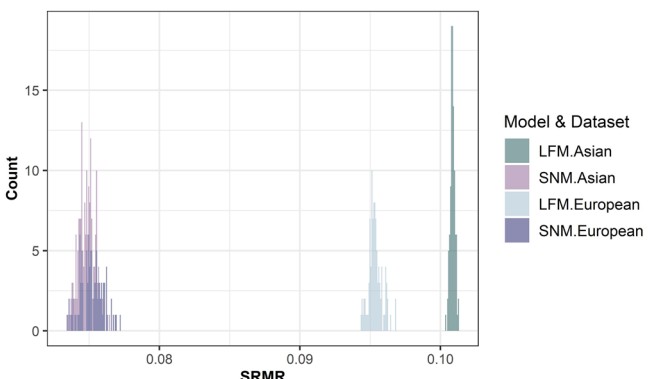

**Fig. 7 | Prediction accuracy of correlations between inferences from two models.** The x-axis indicates the error, i.e., standardized root mean squared residual (SRMR), between the predicted correlations among social inferences based on a specific model (LFM latent factor model (in green), SNM sparse network model (in purple)) and the actual observed correlations among social inferences in each of the samples ($n$ = 100 independent random samples).

CI [27.50, 33.50]. These findings suggest that the high-dimensional network representation better captures the unique correlations between social inferences that were unexplainable by a shared set of latent constructs.

**Network representation captures different links across world regions.** After confirming the validity of network representations for modeling social inferences in samples from different world regions, we next compared the SNMs estimated based on the data from the three samples across Study 1 and Study 2 (Fig. 8). Due to the LASSO regularization in SNM, many partial correlations were shrunk to zero.

We compared the connections between pairwise social inferences in each of the samples in Study 2 to the U.S. sample in Study 1, using the network comparison test. We found many significant differences in the links

of the networks between the U.S. and the Asian sample (Fig. 9A), indicating regional variations in the partial correlations among trait inferences (i.e., correlations between two inferences that were not explainable through other inferences). For instance, the unique correlations between *accent* and *foreign* (difference in $r$ = 0.44, $P$ < 0.001) and between *friendly and happy* (difference in $r$ = 0.22, $P$ < 0.001) were both significantly stronger in the U.S. than Asian participants after accounting for other inferences. We also found a significantly stronger partial correlation between Black and White (difference in $r$ = −0.21, $P$ = 0.040) in the U.S. sample compared to the Asian sample, suggesting that race was more salient in U.S. participants' descriptions, and that the (negative) conceptual association between different racial categories was less explainable through other inferences. We also found significant differences in the partial correlations between social inferences in the U.S. and European samples (Fig. 9B). For instance, after accounting for other inferences, U.S. participants showed a stronger unique link between mentioning a target's *British* origin and describing their *accent* (difference in $r$ = 0.22, $P$ < 0.001), as well as between describing targets as *angry* and *upset* (difference in $r$ = 0.18, $P$ < 0.001).

The overall greater number of unique associations in the US network may reflect more diverse concepts in participants' descriptions compared to Asian and European participants. Word embedding analysis revealed lower average cosine similarity between inferences from the US participants ($M$ = 0.259, SD = 0.024) compared to Asian ($M$ = 0.270, SD = 0.025), $t$ (334202) = −120.71, $P$ < .001, Cohen's $d$ = −0.45, 95% CI [−0.46, −0.45], and European ($M$ = 0.269, SD = 0.025) participants, $t$ (339,498) = −122.81, $P$ < 0.001, Cohen's $d$ = −0.41, 95% CI [−0.42, −0.40]. These results suggest that US participants may draw inferences from a broader conceptual space. This lexical diversity may produce more irreducible partial correlations between inferences. Taken together, these findings highlight how the mental representation of social inferences may be shaped by region-specific cultural experiences.

## Discussion

Across two pre-registered studies, we investigated the mental representation underlying unconstrained, spontaneous social inferences made by a diverse

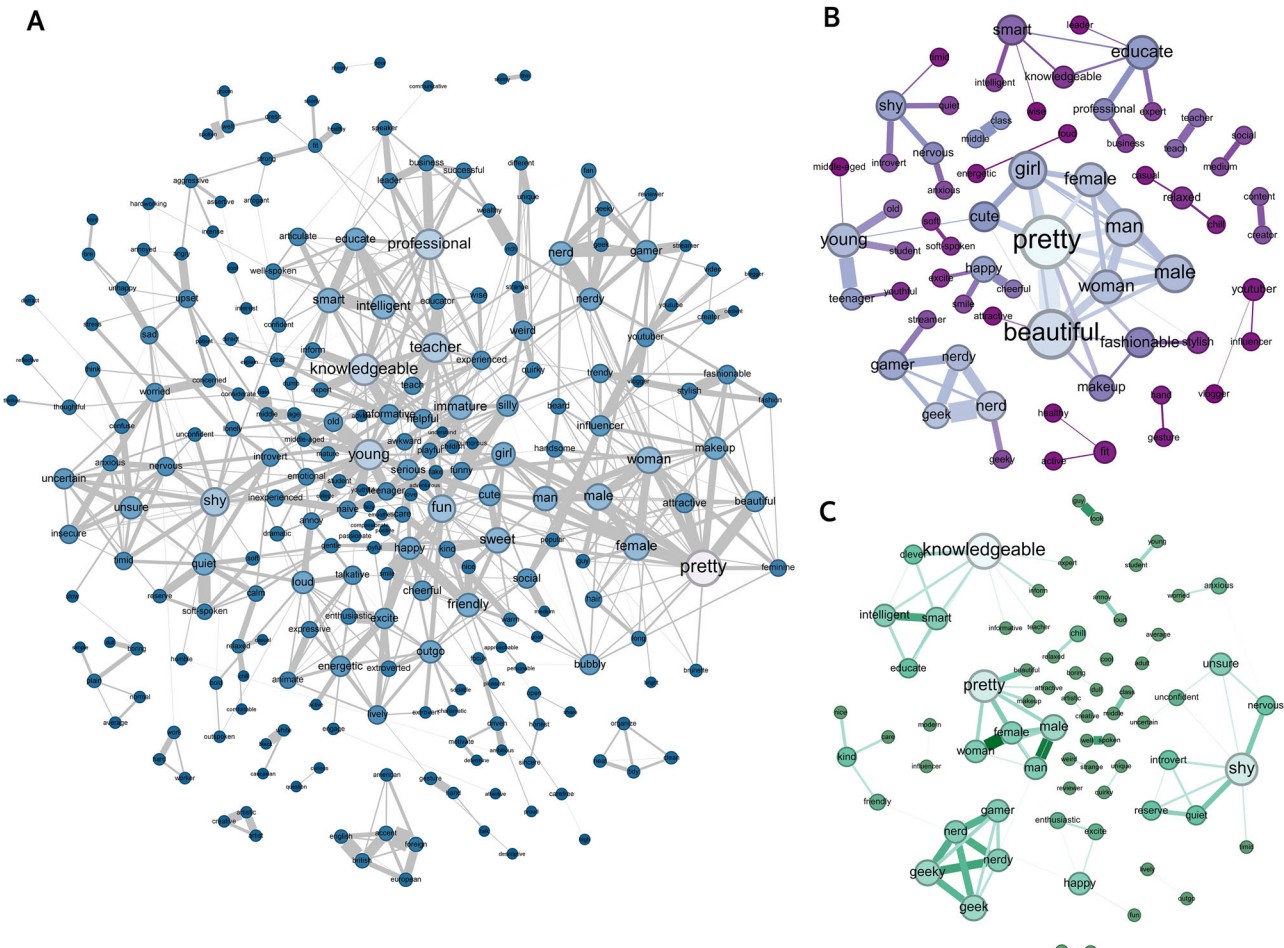

**Fig. 8 | Sparse network models for the U.S., Asian, and Europe samples.** All three networks were estimated using a subset of words that were shared across the three samples and most frequently mentioned by participants ($n = 300$ words). **A** SNM estimated based on the U.S. sample in Study 1. Nodes indicate the social inferences. Edges indicate the partial correlations between two social inferences estimated based on their co-occurrence across videos. The color and size of the nodes indicate the node's strength centrality: lighter color and bigger size indicate greater centrality (nodes with zero centrality were not visualized). Labels for a subset of the nodes with the greatest strength centrality were provided for interpretability. The size of the edges indicates the magnitude of the partial correlation between two social inferences. **B** SNM estimated based on the Asian sample in Study 2. **C** SNM estimated based on the European sample in Study 2.

set of participants based on a representative set of naturalistic videos (Fig. 2, Tables 1 and 2, and Fig. S1). Consistent with prior literature[23–25,41], we found that a relatively large number of latent constructs that were theoretically important (e.g., gender stereotype, physical appearance) explained some of the common variance underlying the relations between naturalistic social inferences (Figs. 1A and 3). However, the amount of common variance explained by latent constructs was much lower than those found in prior literature using more constrained paradigms (e.g., static faces)[3,23,25,41,74]. Instead, we found that a high-dimensional network representation (Fig. 1B), in particular, a sparse network model that encoded the unique, unshared associations between social inferences without assuming shared latent constructs, offered an alternative model that was well-suited for capturing the complex relations between naturalistic social inferences (Fig. 4). We replicated these findings in a U.S. representative sample and two world regions (Figs. 7, 8, S2, and S3): Europe, which is culturally more similar to the U.S., and Asia, which is culturally more distinct[75].

The network models provided valuable insights into the dynamics and diversities of social inferences in naturalistic contexts beyond the latent construct perspective. It identified what social inferences people tended to make together (e.g., shy, depressed) and not make together (e.g., shy, leader; Fig. 5). It informed what social inferences people tended to make based on limited visual information (e.g., physical appearance) and more accumulated multi-modal information (e.g., narrative content, personalities; Fig. 6).

While the low-dimensional latent construct approach often reveals universal structures underlying social inferences across cultures and developmental conditions[23,74,76–80], the network representation revealed significant differences in the unique associations (partial correlations) between social inferences across world regions (Fig. 9). Together, these findings provide an alternative understanding of the structures, dynamics, and diversities of the mental representation underlying social inferences in naturalistic contexts.

Methodological differences may explain why the latent factors struggled to capture common variance in our data. Our experimental paradigm featured much more naturalistic designs than prior research. We showed participants social information that was dynamic, multi-modal, and situated the target individuals in their natural environment (Fig. 2 and Table 1). This paradigm allows for more complex processes through which social inferences were naturally formed[26,28,51,81]. For instance, compared to studies using artificial images, our paradigm revealed additional inferences that were likely driven by the contextual backgrounds (e.g., the inference of "rural") and dynamic information (e.g., the inference of "express"). This highlights the importance of modalities beyond visual cues, such as auditory and semantic information, in shaping social inferences. In contrast, when perceivers are limited to only a small amount of information about the targets (e.g., faces), they are forced to make all inferences requested by the researchers using very similar information available. This may have inflated the correlations between inferences, making their covariance more likely to

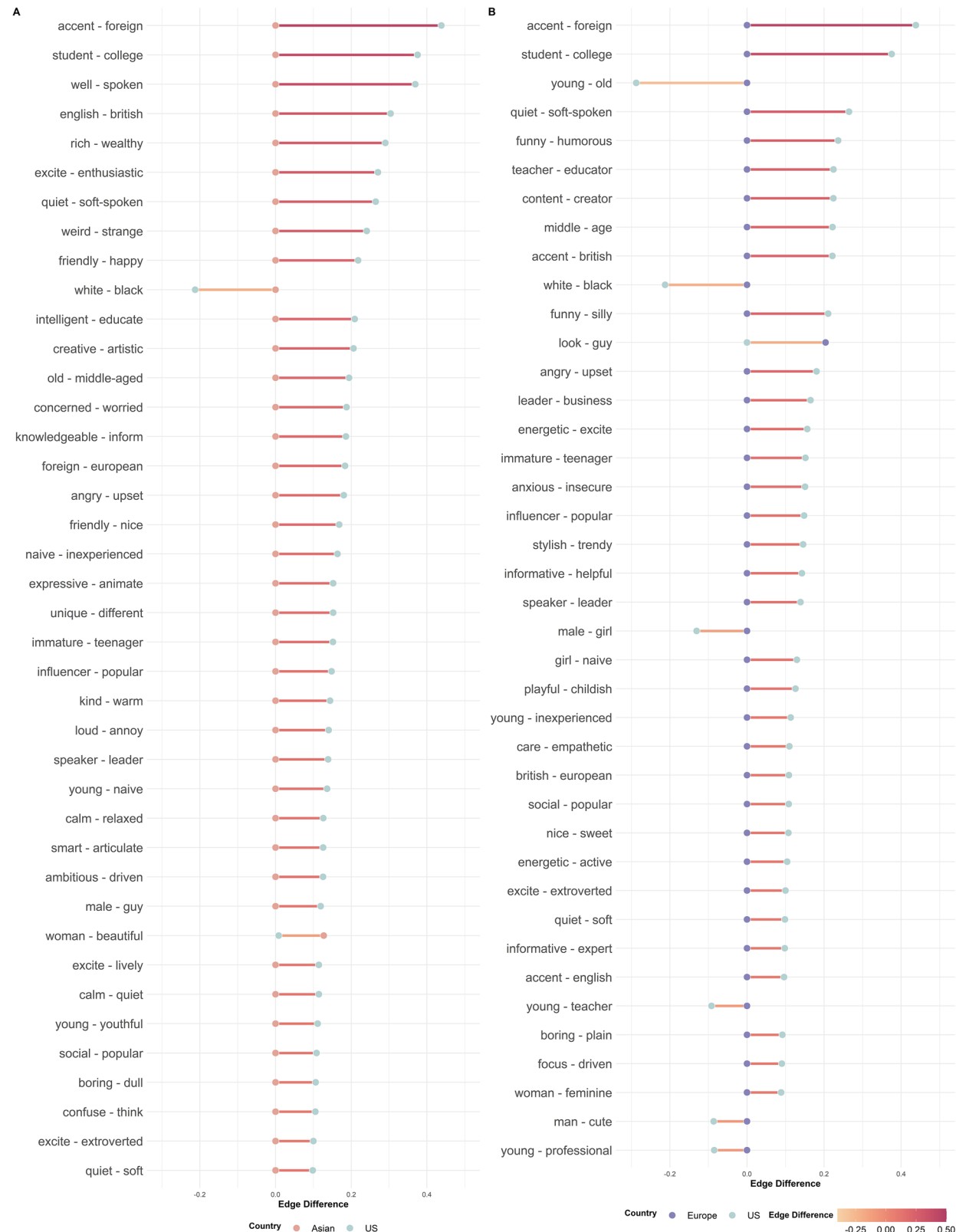

**Fig. 9 | Edge difference between samples from different world regions. A** Top 40 significantly different edges (partial correlations) between the U.S. (cyan dots) and the Asian sample (orange dots). The x-axis indicates the edge weight in each sample. **B** Top 40 significantly different edges between the U.S. (cyan dots) and the European sample (purple dots). All comparisons are based on the same set of inferences ($n = 300$ social inferences).

be explained by a common set of latent factors. However, when provided with a rich set of naturally co-occurring information, perceivers use distinct sets of information to make different social inferences[41]. Social inferences made in this context are not necessarily less correlated (pairwise correlations

between inferences ranged from −0.84 to 0.90), but the pairwise correlations between different subsets of inferences may be driven by distinct mechanisms (e.g., cue use) and thus cannot be explained by a common set of latent factors.

Our findings raise an interesting possibility that the mechanism underlying how people mentally represent social inferences may not be fixed. Both the low-dimensional latent construct representation and the high-dimensional network representation may be plausible depending on the context (e.g., simplified versus complex situations). While the low-dimensional latent construct model offers high interpretability and practicality—a key reason for the substantial and sustained interest it has drawn in the field over the past decades[82], it may lack extensibility and does not capture the full picture. Instead, the latent construct model and the network model may offer complementary hypotheses on how people mentally represent social inferences. The latent construct model focuses on capturing a common set of hidden factors that drive various social inferences (Fig. 1A); whereas, the network model focuses on capturing the unique, (bi-)directional relations between various social inferences (Fig. 1B). These two mechanisms are parallel to the two evolutionary hypotheses of personality[83,84]. One hypothesis, the correlated environment hypothesis, proposes that the correlation between two personality traits arises from the same environmental condition (shared latent constructs) that selects for a high level of both traits. The other hypothesis, the synergistic behaviors hypothesis, suggests that the covariance between two personality traits may be due to individuals high in one trait (e.g., like people) having greater advantages in the other trait (e.g., good social skills). This latter hypothesis also resonates with the hypothesis that human is a dynamic system in which behavioral equilibria are achieved and maintained[33]. Both perspectives may help us understand why some properties of social inferences are shared within a society (e.g., cue usage, stereotypes)[85–87] and why other properties of social inferences (e.g., links between different inferences) differ across societies (Figs. 8 and 9)[37,39].

The results of community detection provide an alternative way to understand latent factors identified in previous studies. Most prior work conceptualizes latent factors as fixed internal forces that cause psychological processes, such as the personality dimensions[84] and the dimensional models of social cognition[3,88]. However, prior work on the latent factors underlying social inferences discovered ever-changing sets of dimensions, from the Big Two[12,86] and Big Three[19,89], to fourteen and even forty dimensions[24,25]. These findings raise the question of whether a fixed set of latent factors underlies social perception and what the relationships are between the latent factors underlying social perception. Our community detection findings offer new insights into these questions. First, we observed latent factor-like structures emerged from the intercorrelations between social inferences in the network (Fig. 5 communities indicated with different colors), such as those related to warmth (purple), competence (magenta), and femininity (blue) found in prior research[23]. Thus, instead of conceptualizing latent factors as fixed internal forces, conceptualizing them as emergent properties of the interconnections between inferences instead may better explain the flexibility of social cognitive dimensions across contexts.

Second, the communities identified based on social inferences (Fig. 5) also bore resemblance to the Big Five personality dimensions. For instance, the orange community (containing inferences such as nerdy and geeky) and the red community (containing inferences such as musician and strange) were similar to the two facets of openness—intellect and esthetic sensitivity; the purple community (containing inferences such as friendly and happy) was similar to agreeableness; and the green community (containing inferences such as depressed and lonely) was similar to neuroticism. This similarity between social inference dimensions and personality dimensions was consistent with prior work showing that perceiver effects in social inferences are structured by the Big Five dimensions[90,91]. This similarity may be explained by the shared lexical foundation of studies on social inferences and those on personality dimensions, which both analyzed verbal descriptors of human behavior[92]. It may also be driven by the bidirectional relationship between self-perception and perception of others: how we perceive others is shaped by our self-perception, and vice versa[93,94].

The network approach demonstrates how a social inference's psychological significance can be understood through its centrality—the extent to which it connects to and influences other judgments within the impression formation process[95]. This perspective is not emphasized in the latent construct representation, where social inferences are interchangeable and only vary in how reliably they measure the latent constructs[33,96]. But the concept of centrality is not new, which was implied in Asch's pioneering work eight decades ago when he investigated the centrality of trait words on overall impressions[14]. His findings already suggested that different traits are not interchangeable but have distinct impacts on other traits. Utilizing the concept of centrality in a network, we showed what different descriptions perceivers tended to write together (Figs. 5 and 8) and what different types of social inferences were central to the descriptions perceivers wrote over time (Fig. 6). This finding suggests that different physical inference information and abstract inference information spreads from one inference to another[70]. Given the assumed flow process of centrality, our results also suggest that biases in impressions formed in earlier processes (e.g., about appearance) may influence those formed in later processes (e.g., personality). Our findings offer new explanations for the mechanisms underlying a wide range of well-known phenomena, such as the beauty-is-good stereotype[47] (attractive people are perceived to have many other positive traits), the coarse-to-fine dynamic impression formation process[26] (from broad social categories to refined individual characteristics), and priming effects of group identities (e.g., gender, age, race) on subsequent judgments[42] (knowing a person's group identity shapes social inferences). However, we caution that the timing of when a description was written down may not perfectly correspond to when the inference was formed in the mind. Nevertheless, empirical studies have shown that the order of written inferences reflects the saliency and discriminability of the inferences, indicating an indirect link between when an inference is formed in the mind and when it was written down by perceivers[2].

## Limitations

Several limitations of our research may constrain the generalizability of our conclusions. First, we tried to elicit as diverse social inferences as possible using naturalistic paradigms. Due to the high-dimensional nature of our data (a lot of spontaneous inferences compared to a relatively smaller number of videos), results from the exploratory factor analysis may be unstable. Nevertheless, empirical studies show that the observation-to-variable ratio (video-to-word ratio) does not affect the stability of factor analysis[97]. Second, although we tried to recruit sufficiently powered and U.S. representative samples, our analyses were conducted at the group level across participants. These findings cannot capture potential variations across perceivers (e.g., different perceivers may have different general tendencies in evaluating other people), which has been shown to be an important factor that shapes social inferences from both naturalistic images[45,98] and naturalistic interpersonal interactions[93,94]. Third, we aim to advance an ecologically valid understanding of social cognition by applying naturalistic paradigms (e.g., naturalistic videos and free responses). However, our findings may not generalize to real-world social interactions where information streams may not follow the same presentational norm as social media videos, and the relationships between targets and perceivers may play an important role[99]. Nevertheless, our naturalistic paradigm and modeling approaches can be naturally extended to analyzing social inferences at the individual level and during social interactions.

## Conclusions

In conclusion, we offer an alternative approach to understanding the mental representation of social inferences, the high-dimensional network approach. We provide initial empirical evidence supporting that this approach is well-suited for capturing complex social inferences in naturalistic contexts. The network approach by no means invalidates the low-dimensional latent construct approach. Instead, it opens new doors for more nuanced and naturalistic investigations of the mechanisms underlying social inferences.

## Data availability

All de-identified data are freely accessible at Open Science Framework: https://osf.io/m42ht/?view_only=e0f3f80f6ace4b05ad2b73c28723a399. All

naturalistic videos and corresponding annotations are from the First Impression video dataset (https://chalearnlap.cvc.uab.cat/dataset/24/description/).

## Code availability

All data were collected via online experiments written in JavaScript. Data were analyzed with Python (3.10) and R (4.4.0) programming language. The code is available at https://osf.io/m42ht/?view_only=e0f3f80f6ace4b05ad2b73c28723a399.

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

## Acknowledgements

The authors received no specific funding for this work. The authors thank other members of the IMPression in ACTion lab for their helpful comments.

## Author contributions

J.L.: conceptualization, methodology, investigation, formal analysis, software, visualization, validation, and writing–original draft. C.L.: conceptualization, supervision, methodology, funding acquisition, writing–review and editing.

## Competing interests

The authors declare no competing interests.
