## [Transparent Peer Review file · Communications Psychology]

Network models reveal high-dimensional social inferences in naturalistic settings beyond latent construct models

Corresponding Author: Mr Junsong Lu

Version 0:

Reviewer comments:

Reviewer #1

(Remarks to the Author)

In this manuscript, the authors tested the dimensionality of spontaneous trait inferences of individuals based on observations of 10-second videos sampled from an online database. They used both a latent variable and a sparse network modelling approach to identify the domains of these spontaneous inferences. They showed the sparse network approach accounted for more variance in perceptions and concluded this approach identified more complex relationships between spontaneous trait inferences. The manuscript is well-written and the authors should be commended for their meticulous efforts in the selection of stimuli and rigorous analytic approach. Below, I outline some concerns that if addressed could increase the contribution of this work to the field.

1. I applaud the authors for investigating spontaneous trait inferences of more naturalistic stimuli. Getting closer to the information available to perceivers during social interactions increases the generality of this work beyond previous studies of spontaneous trait inferences of faces. However, there is an entire field of research on trait impressions in natural social contexts, interpersonal perception, that is not discussed. I believe the manuscript would benefit from engaging with this work and its insights into trait inferences of the most naturalistic stimuli, other people (see Kenny 1994, 2018; Kenny & LaVoie, 1984).

2. I don't think the model of trait perception in Figure 1 accurately represents how latent factors (i.e., personality traits) are related to behaviors and trait inferences. The icons pointed at the latent traits are described as representing "inputs that elicit social inferences such as perceptual inputs of visual, auditory, and speech information." But these inputs cannot be the cause of the latent factors. Instead, the latent factors are the cause of the observable behavior (i.e., perceptual inputs) that are used to make social inferences. For example, the latent construct of extraversion causes individuals to be more talkative and act more assertively. These behaviors are then perceived and used to make inferences about the stable traits of the target. The tendency of these behaviors and inferences to co-occur are then used to organize the inferences into the latent factors, or higher order traits. Unless I misunderstand what the authors mean by inputs, the inputs (information used by the perceiver to make inferences) in the model would be the reflective (observable) indicators of the latent factors, the orange dots (which in latent variable models are typically indicated by rectangles).

3. I am not convinced that the data provide evidence of how social inferences "naturally unfold over time" (p. 6). First, there is no temporal separation between the items used for time 1 and time 2. Sure, the participants listed that item first but there is no way of knowing if that was the first word/inference that came to mind or one they thought was central. Second, it is not clear if the additional inferences were made between time-points 2 and 3, and 3 and 4 based on the 5 seconds of information presented or if people formed their impressions from the still picture and then just continued to list more words when asked. It does not seem possible to distinguish between these possibilities and therefore seems inappropriate to make claims about how social inferences unfold over time based on sets of words listed at three different points within a 10-second video.

4. There is at least one alternative explanation for the additional domains of social inference identified in these stimuli compared to the more static stimuli used in past research. That is, the perceivers were provided with additional information about targets, beyond appearance, speech, and behavior, from the background of the video. It is difficult to tell from the figure, but it appears that each target appeared in a background of their choosing. The additional information from the background might have led perceivers to make additional or different social inferences than if they were presented only the

target. Maybe this information is why there appear to be additional domains than identified in previous work, rather than the informational differences between static and naturalistic stimuli?

5. I appreciate the authors data driven approach and the visualization of the results. Figure 5 is a nice representation of the complexity and organization of spontaneous social inferences. The communities in the network look like they align with big five traits, which would support the centrality of the big five for social perception. The big five comes from words people use to describe each other (e.g., Saucier & Goldberg, 1996), and others have argued that social inferences are organized this way (see Srivastava, 2010). But as the authors note, social perception studies often use a top-down approach that assumes this organization rather than tests it. I am curious if the authors tried to label the communities in Figure 5. It appears there is a gender/attractiveness community (blue), and then one could argue that the nerdy, geeky, technical community and weird, musician strange communities are two facets of openness (intellect and aesthetic sensitivity); the outgoing, friendly, happy community is agreeableness; the quite (quiet; sp?), the lonely, depressed, shy community is neuroticism; the young, knowledgeable, and leader community is extraversion; and the trainer, strong, gym community is competence. I think it is quite remarkable that the authors were able to mostly recover the big five in spontaneous trait perceptions with a network approach. This appears to support the centrality of the big five to trait inferences. The authors came to a different conclusion from the data, so I think it is important for them to address why these communities look like but are not the big five. This pattern also appears in the data from Study 2. In both the Asian and European samples social inferences appear to be organized in big five related domains, which I think is noteworthy and an important point for discussion.

Smaller points:

1. The descriptions of the results below are maybe missing a word or something:

“for instance, in the U.S. sample, when participants thought that the target person was foreign, they were more likely to also thought about the accents of the person than participants in the Asian sample (difference in $r = .44$, $p < .001$)”

“U.S. participants thought that the target person was British, they were more likely to also thought about the accents of the person than participants in the European sample (difference in $r = .22$, $p < .001$)”

2. Figure 9. Why are the white-black dimensions and young-old negative? What does this mean?

3. Figure 7, I am not sure I understand the legend: theory I and theory II are not explained in the text. I assume these relate to EFA versus SNA but maybe could be edited for clarity.

Kenny, D. A., & La Voie. (1984). The Social Relations Model. *Advances in Experimental Social Psychology*, 18, 141-182.

Kenny, D. A. (1994). *Interpersonal perception: A social relations analysis*. Guilford Press.

Kenny, D. A. (2019). *Interpersonal perception: The foundation of social relationships*. Guilford Publications.

Saucier, G., & Goldberg, L. R. (1996). Evidence for the Big Five in analyses of familiar English personality adjectives. *European journal of Personality*, 10(1), 61-77.

Srivastava, S. (2010). The five-factor model describes the structure of social perceptions. *Psychological Inquiry*, 21(1), 69-75.

Reviewer #2

(Remarks to the Author)

The authors present the results of two studies examining the structure of social inference, which argue that our impressions of other people (here, based on free responses to naturalistic stimuli) are better characterized in terms of the network associations between individual inferences than they are by models comprising a handful of specific latent dimensions. They argue that this network approach also captures meaningful cross-cultural differences in terms of associations between specific inferences.

I *really* enjoyed this paper! To be transparent, the core hypotheses are ones that I strongly agree with and that I've been eager for more research to dive into. It's always seemed to me that the relationships between individual trait/emotion/behavior/etc. concepts (and the overarching conceptual map of those associations) are more critical for guiding social inference than the precise number and nomenclature of dimensions that characterize trait/emotion/behavior/etc. space. That said, reading this work sparked a lot of questions for me, which I hope the authors will attend to while revising this work.

1. Early on in the manuscript, I was wondering what information domains the authors were focused on when considering the latent dimensional approach. In other words, does it matter if the inferences being made are specific to traits (or behavioral instances, or facial features, or emotion displays and mental states, or preferences, or group memberships)? As the manuscript unfolded, it became clear that, no, the informational content supporting these inferences doesn't matter – and indeed, in everyday life, we typically make our inferences based on multiple channels of dynamic information perceived

simultaneously.

That said, I think a reader steeped in the dimensional approach (especially as it's been applied to trait inferences and face evaluations) will be a bit surprised to see more complex / more specific inferences (i.e., "gardener," "european," "hippie," "brown-eyed," "punk") alongside more familiar "classic" ones ("friendly," "kind," "intelligent"). The explanation for this approach on pg. 31 was particularly useful and clear for me. I wonder if it might be useful to give some similar commentary earlier on in the paper.

(That said, the detail on pg. 31 is framed, in part, as an explanation of "why the latent constructs struggled to capture most of the common variance in [the authors'] data." I'll admit that I'm having an easier time understanding how their explanation here differentiates the present approach [and findings] from prior work – versus how this explains the difference *between models* in the present work. The same approach was used to generate the inputs to both models [i.e., "we allowed participants to freely write down whatever inferences they thought of, not just trait inferences, nor forcing them to think about a pre-defined set of descriptions"], so it's not clear to me why this design feature explains why the sparse network model fits the data better than the latent dimension model.)

2. When describing their stimulus set (comprising 444 clips taken from a larger set of 3000 YouTube videos), the authors say that "[u]nlike static images and verbal descriptions, naturalistic videos present target individuals through multimodal information streams dynamically, which closely resemble the way people observe each other in real life." It certainly *feels* true that most people spend a considerable amount of time frequently observing other people via social media! That said, I wonder if a related implication of this statement is true – if people's behavior in the videos they post to social media is "like real life." I will reveal my ignorance/inexperience here and say that I've never posted a video to YouTube, but it sure seems like the typical "YouTuber addressing the camera directly" video (across various content domains) is governed by specific presentational norms and conventions.

Some of the descriptions of the factor loadings in Figure 3 (i.e., superficiality, nerd culture, rural lifestyle, tech savviness, entrepreneurial mindset) are adding to my concerns about how broadly representative this stimulus set is, as are some of the social roles depicted in Figure 5 (i.e., gamer, trainer, businessman, musician). Moreover, presumably the participants in this study knew the nature of these stimuli; references to "content creators," "vloggers," "youtubers" in the collected social inferences make this clear. As such, if participants have beliefs/expectations/stereotypes about this particular social category (which many if not most of the targets belong to), could this be shaping and/or constraining the links between individual social inferences?

Regarding the groupings of these videos, just a few additional questions:

- The authors grouped age into two bins – "younger than 33 years old, older than 33 years old" – how was this cutoff determined?
- Similarly, they grouped gender into "female [and] male" and race into "Asian, Black, and White." How were these determined? If automated/algorithmically, how was accuracy assessed and maintained? Assuming that targets' actual self-reported identities were not measured, should these be described as "gender appearance" and "racial appearance"? Does "Asian" collapse across East Asian, South Asian, and Southeast Asian targets?
- The authors state that "The number of videos sampled per group was proportional to the total number of videos of that group in the 1,897 videos." This presumably results in very few videos in some cells (e.g., with 9 videos with Asian targets overall, there would be very few videos in the "Asian men over 33" cell). Why be beholden to that initial proportion – why not aim for greater representation across all possible combinations of age, race, and gender? As it stands, the stimulus set is heavily skewed towards White young adults.

3. One of the aspects of this work that I was most excited to dig into was the cross-cultural comparison between samples, reflecting differences in the association strength between individual inferences. As the authors write early on, "... associations between social inferences may reflect associations between personalities. For instance, people may map social reality of personalities into social perception through cultural learning, so that perceivers in different cultures that are exposed to different personality structures display different mental representations of social inferences." I will say – I think this is a big, important idea that is a *little* underdeveloped in this initial statement. I do think that the authors could go into greater detail here (potentially by expanding on the previous work they're citing) to foreground/scaffold what they're proposing and predicting. (Personally, the framing in terms of "personalities" was also a little narrow – presumably this could also be due to different cultural norms [e.g., regarding interpersonal dynamics, emotion displays, moral foundations, etc.] or linguistic differences in the meaning of specific inference concepts.)

That said, I will say that some of the data related to these cultural comparisons gave me a bit of pause – particularly with regards to their visualization in Figure 9. The framing of these network connections is given on pg. 9 – greater differences between samples reflect cases where one sample co-inferred the traits in question more so than the other sample. For example, "...when U.S. participants thought that the target person was happy, they were more likely to also thought that the person was friendly than participants in the Asian sample." This is a perfect example of what I think this analysis *could* reveal: cases where a given social inference ("happy") leads to an inference of related, but clearly distinct construct ("friendly"). And yet, many of the other examples seem considerably less meaningful – for example, "student – college," "well – spoken," "content – creator," "middle – age," etc. I doubt that it's the case that when U.S. participants infer that a target is "well" (i.e., healthy), they're also more likely to make inferences related to something "spoken" than participants in the Asian sample – or that when U.S. participants infer that a target is "content" (i.e., peaceful, tranquil, etc.), they're also more likely to make inferences related to them being a "creator" than participants in the European sample. These all seem like cases where a given multi-word phrase ("college student," "well spoken," "content creator") is more common in one sample than

another... rather than a difference in the association between separate concepts.

Similarly, I'm not sure how to interpret the effects for pairs that seem like direct synonyms (i.e., "rich – wealthy," "funny – humorous"), words that are used interchangeably (even if they ultimately have slightly different meanings; e.g., "English – British"), or words that are related in that they both describe [separate] categories of specific sociodemographic dimension ("old – middle-aged"). It would sound a bit odd to say, for example, that based on the inference that someone is rich, we might also infer that they are wealthy. The negative correlations are also a bit confusing to me: what does it mean to say that when participants in the US sample thought that a target person was White, they were less likely to also think that the person was Black than participants in the Asian sample? (Or that when participants in the US sample thought that a target person was young, they were relatively less likely to also think that the person was old than participants in the European sample?)

I don't want to lose sight of the big picture here, but these cases did spark some questions for me about what these associations in the sparse network model are actually capturing – i.e., how many of the links in the overall network are like the hard-to-interpret or seemingly artefactual examples above?

4. Finally, I collected a few more minor main text questions that I noted while reading through the manuscript:

- On pg. 4, the authors write, "However, empirical findings comparing the two models have yet to emerge." It's not clear what the antecedent for "the two models" is in this paragraph – is it the latent-dimension approach and the high-dimensional network representation approach?
- On pg. 5, the authors write, "...each social inference and each relation between social inferences may need to be uniquely represented due to the distinct mechanisms underlying these inferences. For instance, different social inferences may be elicited by different perceptual inputs, stereotypes, and motivations." While this makes sense to me, presumably it's *also* the case that the same inference might be supported by very different (and even contradictory inputs). For instance, to use the example given by the authors on pg. 4, one might make the prediction that someone "goes to parties often" based on the (relatively disparate) inputs "likes people," "is single and trying to meet someone," "works as a caterer," and "has a drinking problem" – even though those inputs are not particularly strongly related to each other. If I'm thinking about this correctly, this is a strength of the present approach that previous approaches geared towards identifying latent dimensions don't capture.
- On pg. 8, the authors write, "We first quantified multi-modal features of the targets in each video, including the visual features of facial identity." What does "visual features of facial identity" refer to? I'm assuming it's sociodemographic features like race, gender, and age. If so, how accurate and reliable is this tool?
- Also on pg. 8, the authors state that they "L-2 normalized the features before sampling." I wasn't immediately familiar with the concept of L-2 normalization; what does it refer to?
- On pg. 11, the authors say that they "excluded a participant if they did not meet our inclusion criteria, failed one or more attention checks, or having more than half of the videos excluded because all words across all pauses in those videos were meaningless." This probably isn't a big deal, but I just wanted to note that I stumbled on this sentence a few times, because the concept of "pauses" in the procedures had not yet been explained. (Similarly, at this point, the reader doesn't know what the attention checks are or what a meaningless word would be.)

5. A few additional questions regarding the figures:

- What do the color-coded pentagons in Figure 1B represent?
- What do the inferences "bos" and "ci" in Figure 3 refer to?
- Figure 6 (and the accompanying text on pg. 24) was a little hard for me to grasp. I *think* I'm correctly understanding now that the rows in Figure 6 represent the top five inferences at each time point in terms of change in strength centrality from the previous timepoint. (Therefore, Time1 would just be the five inferences with the highest overall strength centrality?) So the top five at Time1 are "pretty," "bodybuilder," "muscle," "gym," and "trainer;" the top five at Time2 are "man," "woman," "male," "female," and "pretty" again, etc. I think I was tripping on the "total of 19 unique social inferences" aspects of this figure – i.e., it could have been 10 or 20 inferences per time point, but the authors went with 5 per time point for presentation's sake. Is that accurate?

If so, when the authors say, "We found that inferences made at the earlier timepoints were more about physical appearances (e.g., muscular), then about social categories (e.g., gender), followed by descriptions of what the person was talking about (e.g., advise), and finally inferences of abstract personalities (e.g., empower)," these qualitative characterizations are ultimately subjective judgments being made by the authors, right? The trajectory they're describing makes narrative sense, but it seems like it would be just as logical to say that participants first judged whether targets were fitness enthusiasts, then judged their gender identity, then judged their rhetorical skill, and finally judged their power/dominance. I don't mean to be silly; I think my point is just that these interpretations feel a bit overstated, particularly given how these data are presented.

- I don't see a distribution corresponding to the Theory II model of the European dataset in Figure 7. Is it missing? Is it overlapping with the Theory II Asian distribution? If so, is there a way to present this that's a bit clearer, visually?
- The caption for Figure 9 indicates that the figure presents the "Top 10 significantly different edges between the U.S. (cyan dots) and the Asian sample (orange dots)," as well as the top 10 different edges between the U.S. and European samples. However, there 15 rows in each panel of the figure.

6. Finally, I noted a few typos:

- On pg. 4, in "...the network representation explains individual difference in terms of the various causal links between different behavior and cognition," should the last phrase be "different behaviors and cognitions"? (Also, replacing "different" with "specific" might clarify the meaning here.)
- On pg. 7, the authors write, "...we targeted the First Impression video dataset that were gathered by a prior research." I think "by a prior research" should be "in prior research."

- On pg. 11, the authors write, "Since it would take too long for each participant to evaluate each of the 444 videos. We randomly assigned these videos into modules, each with 12 videos." These two sentences should be joined together in one.
- On pg. 31, the authors write, "...when provided with a rich set of information that naturally co-occur, a recent research showed that perceivers use distinct sets of information to form impression of different traits." I'd suggest rephrasing "a rich set of information that naturally co-occur" as "a rich set of naturally co-occurring information," and further, changing "a recent research" to just "recent research."

Reviewer #3

(Remarks to the Author)

In this manuscript, the authors ask whether social inferences are well described by a few latent variables or require more complex models. The authors find that social inferences in response to video clips are poorly explained by latent dimensions, and continue to advance a network model which describes the data better.

In the present studies, participants watched clips and responded freely over multiple timepoints of the clips. The authors also investigated the temporal dynamics of social inferences, and examined cross-cultural differences in network dynamics by sampling US American, Asian, and European participants.

This paper uses an interesting methodological approach to tackle an intriguing problem. I would like to highlight positively the use of open-science methods, the use of naturalistic, complex stimuli, repeated measures, and a network approach. Although I find this paper interesting, I am not sure about the conclusions that can be drawn from the present approach. See my detailed points below.

Major

1. I am not an expert in this particular technique, but 'excluding words that only one participant mentioned and mentioned in less than 1% of the videos', reduces your data from more than 9000 unique words to less than 3000. That seems like an usually large proportion of data, and I also fail to find this in the preregistration (please point me to the relevant section, if it can indeed be found there).
2. Was it actually 'social inference' what people did or just descriptions of these videos? I failed to find the specific instructions used in the prompts, and I wonder whether all these words can be classified as social inference or whether parts of them constituted unrelated comments.
3. Since participants did not structure—by definition—their inferences, it's hard to know which words they thought were central and which other words were peripheral. I suppose that a for instance 2-dimensional model would still posit that peripheral descriptions will revolve around their two dimensions, but nonetheless: Since the authors explicitly gave the option to fill in 10 textboxes, participants may understand that as many as 10 words may be important to the description.
5. You report % variance explained numbers for the 25 and the 100 dimension model, yet I do not find a comparable number based on your network model. I understand that the error of the SNM is lower, but how can I evaluate this quantitatively?
4. Although the idea of studying temporal dynamics in your network model is interesting, it seems rather obvious that surface-level characteristics are described first and content-related characteristics described later. It is good that your model captures this intuition, but I wonder about the contribution of these findings.
5. This may very well reflect my own lack of understanding, but: Could it be that your results are due to autocorrelation between people's inputs? For instance, if participants' inputs are quite consistent but not explained by any underlying factor, a model which suggests low variation in responses will fare well.
6. Please read the manuscript carefully for grammar and typos. E.g. line 171: 'a prior research'. Furthermore, line 242 has a fullstop where it should have a comma, line 277 should say 'groups' instead of 'group', 'experimental paradigm' instead of 'experiment paradigm' on line 701, 'thought' in line 551 should be 'think' (this is a recurring mistake throughout the manuscript), and many more issues. Also, be careful with using 'Figs.' instead of 'Fig.', for instance in line 684.

Minor

1. Please change the R command in line 168 to a verbal description of the procedure.
2. although your figures are visually appealing, they mostly are of low resolution/seem washed out. Could you please export them in higher resolution?

Version 1:

Reviewer comments:

Reviewer #1

(Remarks to the Author)

I appreciate the authors detailed responses to my comments, but I don't think they adequately engaged with some of the broader points. Below, I provide further detail.

1. In their response to Comment #5, the authors miss an opportunity to highlight the broader significance of their findings for the study of impression formation. Instead, they raised a different question about causal mechanisms: "What the relationships between personality dimensions and social cognitive dimensions are is indeed a very interesting question that remains to be addressed. A related and even deeper question is how we should conceptualize these psychological dimensions: are they fixed internal forces or emergent properties?"

I am not sure what the authors are distinguishing here with personality and social cognitive dimensions. If they are referring to warmth and competence, I would say that these too are personality traits but ones that did not emerge as clearly in the present work as the big five. Additionally, while the nature of personality traits—whether biologically determined or emergent—is an ongoing debate, it does not seem central to interpreting the present findings, which suggest the Big Five are central to social inferences. Importantly, the present evidence of the emergence of the Big Five in spontaneous trait inferences was anticipated by prior theoretical work titled: The five-factor model describes the structure of social perceptions (Srivastava, 2010). It also supports decades of research and focus on Big Five impressions in the field of interpersonal perception.

Considering that perceptions made of these more natural dynamic stimuli more closely resemble those made of people rather than static stimuli, the findings also raise an important question about the validity of inferring real-world effects from the study of perceptions of static stimuli.

2. The authors did not address the issue I raised in comment #4. My concern about the background information being an alternative explanation for the additional domains of social perception, compared to static stimuli remains.

3. Finally, I urge the authors to double check that all citations support claims.

- a. the opening sentence states that people make a wide variety of social inferences in daily life but the citations to support this are papers on social inferences from faces (1-4)—not of people in daily life.
- b. Citation 87 and 92 do not appear to support the statement: “Most prior work conceptualizes latent factors as fixed internal forces that cause psychological processes, such as the Big Five personality traits causing individual differences in behavior” (87,92).”
- c. Citation 93 does not support a 3 dimensional model—it is a paper validating the Big Five—“The results are straightforward, showing convergent and discriminant cross-observer and cross instrument validation for all five factors.” (McCrae & Costa, 1987, p. 86)
- i. The other citation for a three-factor model of social inferences (19) comes from the study of stereotype content of social group labels, not inferences about individuals.

Reviewer #2

(Remarks to the Author)

The authors' revision was very responsive to the questions and critiques raised in the first round of review. In particular, I found their response letter to be particularly comprehensive and informative. From my perspective, the manuscript has been usefully clarified and improved. I have a few remaining stray thoughts/questions that I've listed below, which I hope the authors will consider as they move forward. Thank you once again for the opportunity to consider this work!

1. In their response to my comments, the authors posited a plausible account of why some of the cross-cultural differences may have emerged in Study 2. As they write, “...participants from the U.S. have a bigger English vocabulary for describing the same concept using different words than Asian and European participants. This is likely because our U.S. participants were required to be native English speakers, but our Asian and European participants were only required to be fluent in English.” I don't know that I'd fully processed how that difference between samples might impact the observed network representations. It's not a major issue, but if the authors do indeed find this to be a reasonable explanation, it's something worth drawing the reader's attention to more explicitly. (More generally, I hadn't considered how these differences in correlations could reflect both a) an increased association between concepts in one sample vs. another and b) increased discussion overall of a given topic in one sample vs. another, so the authors' explanation was helpful.)

That said, I wanted to try to estimate the magnitude of the difference the authors are describing here in terms of vocabulary/usage differences across samples. The authors state that after processing/exclusion, there were 2964 unique words used to describe the Study 1 videos, which, with 1598 participants comes out to 1.85 unique words per participant. If what the authors are proposing is accurate, the number of unique words per participant should be lower in both the Study 2 Asian and European samples. By this metric, it is in the European sample ($1319/792 = 1.66$ unique words per participant), but not in the Asian sample ($1351/651 = 2.07$ unique words per participant). That said, maybe I'm not thinking about this the right way and/or this isn't the best metric of vocabulary (or if differences of this magnitude on this metric are meaningful). In any case, it would be helpful to get the authors' take on this.

2. On the subject of the descriptions/inferences generated spontaneously by participants, it strikes me that (as in other related work like Connor et al., 2024), the descriptions given are on the whole, somewhat positively skewed (i.e., I'm not seeing a lot of “untrustworthy,” “ugly,” “dumb,” “weak,” etc.). Beyond that, while participants are making some inferences regarding character traits (i.e., things like intelligent and kind appear among the inferences in Figure 3), there are far more inferences about stable aspects of identity (i.e., age, gender, etc.), social roles (i.e., farmer, businessman), and emotionality (i.e., upset, angry, happy). Is there any way of quantifying the valence skew (either overall or within the most trait-like words) in the set of inferences? Moreover, if the set is positively skewed, is there any concern that this reflects some degree of social desirability bias? (Perhaps participants are inferring certain features that they don't feel it's appropriate to report on.)

3. When reading Point #4 from Reviewer 1 (and the authors' response), I got to thinking about the other social cues that are present in the videos (i.e., cues to social class, body shape and size, environmental context, etc.) that were not considered in the feature quantification / extraction step described on pages 9 and 10. (The Vikings shirt that Bean Woman in Fig. 2A is

wearing is an important cue for me, personally!) Just thinking this through... what's the consequence of not characterizing something like social class and accounting for it in the maximum variation sampling? Is it just that there might be less variation on that feature across the full stimulus set than there is for the features described on page 9?

4. Finally, I just had a small wording note. On pg. 36, the authors write, "The network representation also highlights the importance of centrality in understanding social inferences. That is, the importance of a social inference in terms of its connections to other social inferences." The second sentence here is a fragment – can these two thoughts be joined?

Reviewer #3

(Remarks to the Author)

The authors have addressed my questions thoughtfully and to my satisfaction.

Version 2:

Reviewer comments:

Reviewer #1

(Remarks to the Author)

I appreciate the author's responsiveness and engagement. In the current manuscript, the authors have adequately addressed my previous comments and made important connections with the broader literature. My thanks to you and the authors for asking me to review this manuscript.

Reviewer #2

(Remarks to the Author)

The authors have addressed all of my remaining questions and comments. Thank you again for your careful and comprehensive revision of this manuscript -- and for the opportunity to read this work!

Reviews

Reviewer 1

REVIEWER EXPERTISE:

Reviewer #1: Computational modeling, person perception

Reviewer #2: Person/social perception

Reviewer #3: Computational modeling, person perception

REVIEWER REPORTS:

Reviewer #1 (Remarks to the Author):

In this manuscript, the authors tested the dimensionality of spontaneous trait inferences of individuals based on observations of 10-second videos sampled from an online database. They used both a latent variable and a sparse network modelling approach to identify the domains of these spontaneous inferences. They showed the sparse network approach accounted for more variance in perceptions and concluded this approach identified more complex relationships between spontaneous trait inferences. The manuscript is well-written and the authors should be commended for their meticulous efforts in the selection of stimuli and rigorous analytic approach. Below, I outline some concerns that if addressed could increase the contribution of this work to the field.

1. I applaud the authors for investigating spontaneous trait inferences of more naturalistic stimuli. Getting closer to the information available to perceivers during social interactions increases the generality of this work beyond previous studies of spontaneous trait inferences of faces. However, there is an entire field of research on trait impressions in natural social contexts, interpersonal perception, that is not discussed. I believe the manuscript would benefit from engaging with this work and its insights into trait inferences of the most naturalistic stimuli, other people (see Kenny 1994, 2018; Kenny & LaVoie, 1984).

RESPONSE: Thank you for this positive assessment and for pointing us to the relevant literature on naturalistic interpersonal perception. We have now included this literature in the revised Discussion:

“Second, although we tried to recruit sufficiently powered and U.S. representative samples, our analyses were conducted at the group level across participants. These findings cannot capture potential variations across perceivers (e.g., different perceivers may have different general tendencies in evaluating other people), which has been shown to be an important factor that shapes social inferences from both naturalistic images^{47,99} and naturalistic interpersonal interactions^{100,101}. Third, we aim to advance an ecologically valid understanding of social cognition by applying naturalistic paradigms. However, our findings may not generalize to real-world social interactions where the relationships between targets and perceivers may play an important role¹⁰². Nevertheless, our naturalistic paradigm and modeling approaches can be naturally extended to analyzing social inferences at the individual level and during social interactions.”

2. I don't think the model of trait perception in Figure 1 accurately represents how latent factors (i.e., personality traits) are related to behaviors and trait inferences. The icons pointed at the latent traits are described as representing "inputs that elicit social inferences such as perceptual inputs of visual, auditory, and speech information." But these inputs cannot be the cause of the latent factors. Instead, the latent factors are the cause of the observable behavior (i.e., perceptual inputs) that are used to make social inferences. For example, the latent construct of extraversion causes individuals to be more talkative and act more assertively. These behaviors are then perceived and used to make inferences about the stable traits of the target. The tendency of these behaviors and inferences to co-occur are then used to organize the inferences into the latent factors, or higher order traits. Unless I misunderstand what the authors mean by inputs, the inputs (information used by the perceiver to make inferences) in the model would be the reflective (observable) indicators of the latent factors, the orange dots (which in latent variable models are typically indicated by rectangles).

RESPONSE: We thank the reviewer for pointing out this confusion. We would like to clarify that the latent factor model in Figure 1 describes the mental model of social perception, where the latent factors correspond to core social inferences (e.g., the Stereotype Content model dimensions). By analogy, for color perception, perceptual cues are first encoded along the three color dimensions (red, green, and blue), whose combination then results in the perception of specific colors. This is different from the latent model of individual differences, where the latent factors correspond to personality traits (e.g., the Big Five traits) that cause behavior. While it is plausible that these two mental models, one for perception and the other for the ground-truth of personality, may be highly related, there has yet been empirical evidence demonstrating the correspondence between the two. Our study focuses on social perception without getting into the dynamics between perceivers and targets, which we agree would be an important future direction as mentioned in our response to comment #1. But here, the latent factors in Figure 1 reflect how perceivers may organize their inferences of target people rather than how the behavior of the targets in the videos may be explained by their personality traits. Therefore, the links from perceptual inputs to the latent factors do not indicate causal explanation of personality traits but rather a potential causal process of social perception – perceptual cues activate specific dimensions whose combination drive specific social inferences. To minimize confusion, we have now revised this figure to remove the perceptual inputs, since the relations between perceptual inputs and latent factors / inferences are not the focus of our paper. We have also clarified in the revised legend that the underlying structure of social inferences may be different from that of personality traits.

Figure 1. Two models describing mental representations of social inferences. (A) The low-dimensional latent construct model. Orange dots indicate social inferences. Blue dots indicate latent dimensions. Straight lines indicate the associations between social inferences and latent dimensions. Broken curves indicate the associations between latent dimensions. (B) The high-dimensional network model. Orange dots indicate social inferences, and lines (e.g., the highlighted blue line) indicate unique associations between them. Note that both models describe the underlying psychological structure of social inferences instead of personality traits since perceivers' inferences may not reflect targets' true characteristics.

3. I am not convinced that the data provide evidence of how social inferences “naturally unfold over time” (p. 6). First, there is no temporal separation between the items used for time 1 and time 2. Sure, the participants listed that item first but there is no way of knowing if that was the first word/inference that came to mind or one they thought was central. Second, it is not clear if the additional inferences were made between time-points 2 and 3, and 3 and 4 based on the 5 seconds of information presented or if people formed their impressions from the still picture and then just continued to list more words when asked. It does not seem possible to distinguish between these possibilities and therefore seems inappropriate to make claims about how social inferences unfold over time based on sets of words listed at three different points within a 10-second video.

RESPONSE: Thank you for raising this important point about the temporal dynamics of social inferences. We agree that the lack of explicit temporal separation between the items listed at different time points is a limitation of our study. The order in which participants wrote down the items reflects their decisions about what to write and when to write but may not necessarily reflect the order of inferences formed in their minds. We have now removed misleading language when describing the Results, used more neutral language when describing the implications of these results, and mentioned this limitation in the Discussion (attached below). However, we would also like to highlight that using the free response paradigm to investigate and make inferences about the underlying psychological process of person perception has long been a popular paradigm in the field (e.g., Susan Fiske & Martha Cox, 1979). Empirical studies have provided evidence supporting that the order in which descriptors were written in a free response paradigm does carry information about the saliency and ease of the corresponding inferences in the mind (e.g., Lavan, 2023). For instance, Lavan (2023) showed that the more different orders in which an inference was written for two different stimuli in a free response task, the more discriminable the two stimuli were in terms of that inference in a discrimination task. Thus, the order in which descriptors were written down in a free response task does reflect information relevant to the process of social inferences, even though higher-level decision-making (e.g., choosing what words to write) may have also shaped the order.

Revised Results: “The network representation not only revealed the dynamics of social perception in terms of what descriptions people wrote together regarding a target but also what descriptions people wrote over time. We estimated an SNM for inferences entered at four different time points: the first word participants wrote based on the first frame of the video, additional words they wrote based on the first frame, the words they wrote after viewing half of the video, and the words they wrote after viewing the entire video. For each network, we computed the strength centrality of each inference, which is the sum of all edges

connecting that inference to other inferences. Using these strength centralities, we assessed the change in connections between social inference across the four timepoints (Fig. 6). We found a greater increase in strength centrality of physical appearances/fitness (e.g., muscular) and gender identity (e.g., female) from timepoint 1 to timepoint 2, of speech content/rhetorical skill (e.g., mentor) from timepoint 2 to timepoint 3, and of abstract personalities/power (e.g., executive) from timepoint 3 to timepoint 4. These findings suggest that the social inferences that were central to generating other new inferences changed over time, shifting from more physical inferences to more abstract inferences. Moreover, physical inferences continued to show high strength centrality across all four time points, suggesting their sustained influence on the descriptions perceivers wrote across all time points.”

Revised Implications: “These findings suggest that the social inferences that were central to generating other new inferences changed over time, shifting from more physical inferences to more abstract inferences. Moreover, physical inferences continued to show high strength centrality across all four time points, suggesting their sustained influence on the descriptions perceivers wrote across all time points.”

Revised Discussion: “Utilizing the concept of centrality in a network, we showed what different descriptions perceivers tended to write together (Figs. 5 & 8) and what different types of social inferences were central to the descriptions perceivers wrote over time (Fig. 6). This finding suggests that different physical inference information and abstract inference information spreads from one inference to another⁷⁰. Given the assumed flow process of centrality, our results also suggest that biases in impressions formed in earlier processes (e.g., about appearance) may influence those formed in later processes (e.g., personality). Our findings offer new explanations for the mechanisms underlying a wide range of well-known phenomena, such as the beauty-is-good stereotype⁴⁶ (attractive people are perceived to have many other positive traits), the coarse-to-fine dynamic impression formation process²⁶ (from broad social categories to refined individual characteristics), and priming effects of group identities (e.g., gender, age, race) on subsequent judgments⁴¹ (knowing a person’s group identity shapes social inferences). However, we caution that the timing of when a description was written down may not perfectly correspond to when the inference was formed in the mind. Nevertheless, empirical studies have shown that the order of written inferences reflects the saliency and discriminability of the inferences, indicating an indirect link between when an inference is formed in the mind and when it was written down by perceivers².”

Lavan, N. (2023). How do we describe other people from voices and faces? *Cognition*, 230, 105253.

4. There is at least one alternative explanation for the additional domains of social inference identified in these stimuli compared to the more static stimuli used in past research. That is, the perceivers were provided with additional information about targets, beyond appearance, speech, and behavior, from the background of the video. It is difficult to tell from the figure, but it appears that each target appeared in a background of their choosing. The additional information from the background might have led perceivers to make additional or different social inferences than if they were presented only the target. Maybe this information is why there appear to be additional domains than identified in previous work, rather than the informational differences between static and naturalistic stimuli?

RESPONSE: Thank you for this insightful observation. Yes, targets in the videos appeared in a background of their choosing. These backgrounds provide contextual information that may shape perceivers' inferences. In fact, consistent with the reviewer's prediction, a recent study using static images with diverse backgrounds found a larger variety of social inferences (Connor et al., 2024). However, the naturalistic and dynamic stimuli in our study still adds additional complexity to the domains and mental representations of social inferences. For instance, we identified factors such as dominance, where top words included *loud*, *express*, and *energize*. These descriptions suggest that auditory information, beyond solely static contextual information, played a significant role in shaping social inferences.

5. I appreciate the authors data driven approach and the visualization of the results. Figure 5 is a nice representation of the complexity and organization of spontaneous social inferences. The communities in the network look like they align with big five traits, which would support the centrality of the big five for social perception. The big five comes from words people use to describe each other (e.g., Saucier & Goldberg, 1996), and others have argued that social inferences are organized this way (see Srivastava, 2010). But as the authors note, social perception studies often use a top-down approach that assumes this organization rather than tests it. I am curious if the authors tried to label the communities in Figure 5. It appears there is a gender/attractiveness community (blue), and then one could argue that the nerdy, geeky, technical community and weird, musician strange communities are two facets of openness (intellect and aesthetic sensitivity); the outgoing, friendly, happy community is agreeableness; the quiet (quiet; sp?), the lonely, depressed, shy community is neuroticism; the young, knowledgeable, and leader community is extraversion; and the trainer, strong, gym community is competence. I think it is quite remarkable that the authors were able to mostly recover the big five in spontaneous trait perceptions with a network approach. This appears to support the centrality of the big five to trait inferences. The authors came to a different conclusion from the data, so I think it is important for them to address why these communities look like but are not the big five. This pattern also appears in the data from Study 2. In both the Asian and European samples social inferences appear to be organized in big five related domains, which I think is noteworthy and an important point for discussion.

RESPONSE: Thank you for your insightful feedback. What the relationships between personality dimensions and social cognitive dimensions are is indeed a very interesting question that remains to be addressed. A related and even deeper question is how we should conceptualize these psychological dimensions: are they fixed internal forces or emergent properties? We have now incorporated the reviewer's suggestion to provide new insights into these questions using our community detection results. In the revised Discussion, we have added that, "The results of community detection provide an alternative way to understanding latent factors identified in previous studies. Most prior work conceptualizes latent factors as fixed internal forces that cause psychological processes, such as the Big Five personality traits causing individual differences in behavior^{87,92}. However, prior work on the latent factors underlying social inferences discovered ever changing sets of dimensions, from the Big Two^{12,89} and Big Three^{19,93}, to fourteen and even forty dimensions^{24,25}. These findings raise the questions of whether a fixed set of latent factors underlies social perception and what the relationships are between latent factors underlying personality and those underlying social perception. Our community detection findings offer new insights into these questions. First, we observed latent-factor-like structures emerged from the intercorrelations between

social inferences in the network (Fig. 5 communities indicated with different colors), such as those related to warmth (purple), competence (magenta), and femininity (blue) found in prior research¹. Thus, instead of conceptualizing latent factors as fixed internal forces, conceptualizing them as emergent properties of the interconnections between inferences instead may better explain the flexibility of social cognitive dimensions across contexts. Second, these communities not only resemble latent factors previously found in social perception research but also those found in personality research, such as neuroticism (green) and agreeableness (purple). These findings suggest that there may be overlaps between the psychological dimensions of personality and those of social perception⁹².”

Smaller points:

1. The descriptions of the results below are maybe missing a word or something:

“for instance, in the U.S. sample, when participants thought that the target person was foreign, they were more likely to also thought about the accents of the person than participants in the Asian sample (difference in $r = .44$, $p < .001$)”

“U.S. participants thought that the target person was British, they were more likely to also thought about the accents of the person than participants in the European sample (difference in $r = .22$, $p < .001$)”

RESPONSE: Thanks for pointing this out. We have now revised these sentences for better clarity. “We compared the connections between pairwise social inferences in each of the samples in Study 2 to the U.S. sample in Study 1, using the network comparison test. We found many significant differences in the links in the networks between the U.S. and the Asian sample (Fig. 9A). For instance, U.S. participants were more likely than Asian participants to write about the target’s accent when they mentioned that the target was foreign (difference in $r = .44$, $p < .001$), and to write that the target was friendly when they mentioned that the target was happy (difference in $r = .22$, $p < .001$). We also found a significantly stronger negative association between Black and White (difference in $r = -.21$, $p = .040$) in the U.S. sample compared to the Asian sample, suggesting that U.S. participants often wrote about the race of the targets, resulting in a negative correlation between these two descriptions in U.S. participants. This stronger negative association in the U.S. may reflect how racial categories are highly salient concepts for social perception in the U.S. Significant differences in the interconnections between social inferences were also observed between the U.S. and European samples (Fig. 9B). Compared to European participants, U.S. participants were more likely to write about the target's accent when they mentioned that the target was British (difference in $r = .22$, $p < .001$), and to write that the target was upset when they mentioned that the target was angry (difference in $r = .18$, $p < .001$).”

2. Figure 9. Why are the white-black dimensions and young-old negative? What does this mean?

RESPONSE: The stronger and negative associations in the U.S. indicate that the pair of descriptions were more often written by U.S. participants and that the two descriptions were antonyms (i.e., when participants described a target as “White”, it was less likely that they also described the target as “Black”; when participants described a target as “young”, it was

less likely that they also described the target as “old”). We have now clarified these interpretations in the revised manuscript. “We also found a significantly stronger negative association between Black and White (difference in $r = -.21$, $p = .040$) in the U.S. sample compared to the Asian sample, suggesting that U.S. participants often wrote about the race of the targets, resulting in a negative correlation between these two descriptions from U.S. participants. This stronger negative association in the U.S. may reflect how racial categories are highly salient concepts for social perception in the U.S.”

3. Figure 7, I am not sure I understand the legend: theory I and theory II are not explained in the text. I assume these relate to EFA versus SNA but maybe could be edited for clarity.

RESPONSE: Thanks for pointing this out. We have now revised the legend accordingly, using the term “LFM” to indicate latent factor model and “SNM” to indicate sparse network model.

Figure 7. Prediction accuracy of correlations between inferences from two models. The x-axis indicates the error, i.e., standardized root mean squared residual (SRMR), between the predicted correlations among social inferences based on a specific model (LFM: latent factor model, in green; SNM: sparse network model, in purple) and the actual observed correlations among social inferences in each of the samples.

Kenny, D. A., & La Voie. (1984). The Social Relations Model. *Advances in Experimental Social Psychology*, 18, 141-182.

Kenny, D. A. (1994). *Interpersonal perception: A social relations analysis*. Guilford Press.

Kenny, D. A. (2019). *Interpersonal perception: The foundation of social relationships*. Guilford Publications.

Saucier, G., & Goldberg, L. R. (1996). Evidence for the Big Five in analyses of familiar English personality adjectives. *European journal of Personality*, 10(1), 61-77.

Srivastava, S. (2010). The five-factor model describes the structure of social perceptions. *Psychological Inquiry*, 21(1), 69-75.

Reviewer 2

Reviewer #2 (Remarks to the Author):

The authors present the results of two studies examining the structure of social inference, which argue that our impressions of other people (here, based on free responses to naturalistic stimuli) are better characterized in terms of the network associations between individual inferences than they are by models comprising a handful of specific latent dimensions. They argue that this network approach also captures meaningful cross-cultural differences in terms of associations between specific inferences.

I *really* enjoyed this paper! To be transparent, the core hypotheses are ones that I strongly agree with and that I've been eager for more research to dive into. It's always seemed to me that the relationships between individual trait/emotion/behavior/etc. concepts (and the overarching conceptual map of those associations) are more critical for guiding social inference than the precise number and nomenclature of dimensions that characterize trait/emotion/behavior/etc. space. That said, reading this work sparked a lot of questions for me, which I hope the authors will attend to while revising this work.

1. Early on in the manuscript, I was wondering what information domains the authors were focused on when considering the latent dimensional approach. In other words, does it matter if the inferences being made are specific to traits (or behavioral instances, or facial features, or emotion displays and mental states, or preferences, or group memberships)? As the manuscript unfolded, it became clear that, no, the informational content supporting these inferences doesn't matter – and indeed, in everyday life, we typically make our inferences based on multiple channels of dynamic information perceived simultaneously.

That said, I think a reader steeped in the dimensional approach (especially as it's been applied to trait inferences and face evaluations) will be a bit surprised to see more complex / more specific inferences (i.e., “gardener,” “european,” “hippie,” “brown-eyed,” “punk”) alongside more familiar “classic” ones (“friendly,” “kind,” “intelligent”). The explanation for this approach on pg. 31 was particularly useful and clear for me. I wonder if it might be useful to give some similar commentary earlier on in the paper.

(That said, the detail on pg. 31 is framed, in part, as an explanation of “why the latent constructs struggled to capture most of the common variance in [the authors'] data.” I'll admit that I'm having an easier time understanding how their explanation here differentiates the present approach [and findings] from prior work – versus how this explains the difference *between models* in the present work. The same approach was used to generate the inputs to both models [i.e., “we allowed participants to freely write down whatever inferences they thought of, not just trait inferences, nor forcing them to think about a pre-defined set of

descriptions”], so it’s not clear to me why this design feature explains why the sparse network model fits the data better than the latent dimension model.)

RESPONSE: We thank the reviewer for this positive assessment. We have now revised our Introduction and Discussion to clarify why given our naturalistic designs used to generate the inputs, the sparse network model would fit the inputs better than the latent factor model.

In the Introduction, we first introduced the low-dimensional latent factor perspective and pointed out that the ever-expanding latent dimensions found in recent work using more naturalistic designs questions the validity of the latent factor model for capturing social inferences in naturalistic contexts. We then introduced the high-dimensional network perspective, where we have now explicitly addressed why this high-dimensional network representation may better capture social inferences in naturalistic contexts.

“Such high-dimensional network representations may also capture mental representations of social inferences in naturalistic contexts better than the low-dimensional latent factor representations. First, in the real world, associations between social inferences may reflect associations between personalities. This is because people may rely on social reality of personalities to inform social perception through cultural learning. For instance, exposure to different personality structures in different cultures shapes the way people conceptualize trait associations, which in turn guide social inferences^{38,39}. Thus, if the network representation better captures personality, it would also capture social inferences. More broadly, this cultural learning perspective to social inferences also predicts that other factors beyond personality, such as cultural norm and language, would lead to significant cultural variations in the network representations of social inferences^{40,41}.

Second, the artificial designs used in prior research may have inflated the covariance among social inferences and thus favored the low-dimensional latent factor representation. For instance, when participants were forced to rate the targets on a range of traits based on their faces alone, these different trait inferences may rely on very similar and limited cues from the face and thus inflating the correlations between trait inferences. However, in the real world, people observe a wealth of information streams of other people, and they use distinct cues to make different inferences⁴². Thus, the degree to which different social inferences in naturalistic contexts commonly covary – the assumption of the latent factor model – may be reduced. More broadly, different social inferences in naturalistic contexts may be elicited by different perceptual inputs, stereotypes, and motivations^{26,28,42–47}. Social inferences may also elicit one another through heuristics (perceived attractiveness elicits perception of other positive characteristics such as intelligence^{48–51}) and other conceptual biases (stereotypical relations between feminine-looking individuals and characteristics traditionally associated with females^{46,52}). Thus, the complex relations between social inferences in naturalistic contexts may be better captured by high-dimensional network representations, which allow for unique variations across inferences, than the latent factor representation, which assumes a large amount of common variation across inferences.”

In the revised Discussion, we have also further explained why the sparse network model better captured social inferences in naturalistic contexts than the latent factor model. “Methodological differences may explain why the latent factors struggled to capture common variance in our data. Our experimental paradigm featured much more naturalistic designs

than prior research. We showed participants social information that was dynamic, multimodal, and situated the target individuals in their natural environment (Fig. 2; Table 1). This paradigm allows for more complex processes through which social inferences were naturally formed^{26,28,52,83}. In contrast, when perceivers are limited to only a small amount of information about the targets (e.g., faces), they are forced to make all inferences requested by the researchers using very similar information available. This may have inflated the correlations between inferences, making their covariance more likely to be explained by a common set of latent factors. However, when provided with a rich set of naturally co-occurring information, perceivers use distinct sets of information to make different social inferences⁴². Social inferences made in this context are not necessarily less correlated (pairwise correlations between inferences ranged from -0.84 to 0.90), but that the pairwise correlations between different subsets of inferences may be driven by distinct mechanisms (e.g., cue use) and thus cannot be explained by a common set of latent factors.”

2. When describing their stimulus set (comprising 444 clips taken from a larger set of 3000 YouTube videos), the authors say that “

unlike static images and verbal descriptions, naturalistic videos present target individuals through multimodal information streams dynamically, which closely resemble the way people observe each other in real life.” It certainly *feels* true that most people spend a considerable amount of time frequently observing other people via social media! That said, I wonder if a related implication of this statement is true – if people’s behavior in the videos they post to social media is “like real life.” I will reveal my ignorance/inexperience here and say that I’ve never posted a video to YouTube, but it sure seems like the typical “YouTuber addressing the camera directly” video (across various content domains) is governed by specific presentational norms and conventions.

Some of the descriptions of the factor loadings in Figure 3 (i.e., superficiality, nerd culture, rural lifestyle, tech savviness, entrepreneurial mindset) are adding to my concerns about how broadly representative this stimulus set is, as are some of the social roles depicted in Figure 5 (i.e., gamer, trainer, businessman, musician). Moreover, presumably the participants in this study knew the nature of these stimuli; references to “content creators,” “vloggers,” “youtubers” in the collected social inferences make this clear. As such, if participants have beliefs/expectations/stereotypes about this particular social category (which many if not most of the targets belong to), could this be shaping and/or constraining the links between individual social inferences?

RESPONSE: Thank you for your thoughtful feedback. Your point is well taken. These YouTube videos bring us one step closer to understanding naturalistic person perception because they present a wealth of naturally co-occurring information dynamically as how people typically observe others in daily life. However, it is true that YouTube videos may follow specific presentational norms and conventions. Even though our participants were not explicitly told that these were videos from YouTube, they may still be able to make such inferences. As discussed in our paper, top-down influences like this may shape social inferences made. We have now highlighted this limitation in our revised Discussion. “Third, we aim to advance an ecologically valid understanding of social cognition by applying naturalistic paradigms (e.g., naturalistic videos and free responses). However, our findings may not generalize to real-world social interactions where information streams may not

follow the same presentational norm as social media videos and the relationships between targets and perceivers may play an important role¹⁰¹. Nevertheless, our naturalistic paradigm and modeling approaches can be naturally extended to analyzing social inferences at the individual level and during social interactions.”

3. Regarding the groupings of these videos, just a few additional questions:
 - The authors grouped age into two bins – “younger than 33 years old, older than 33 years old” – how was this cutoff determined?
 - Similarly, they grouped gender into “female [and] male” and race into “Asian, Black, and White.” How were these determined? If automated/algorithmically, how was accuracy assessed and maintained? Assuming that targets’ actual self-reported identities were not measured, should these be described as “gender appearance” and “racial appearance”? Does “Asian” collapse across East Asian, South Asian, and Southeast Asian targets?
 - The authors state that “The number of videos sampled per group was proportional to the total number of videos of that group in the 1,897 videos.” This presumably results in very few videos in some cells (e.g., with 9 videos with Asian targets overall, there would be very few videos in the “Asian men over 33” cell). Why be beholden to that initial proportion – why not aim for greater representation across all possible combinations of age, race, and gender? As it stands, the stimulus set is heavily skewed towards White young adults.

RESPONSE: These are good questions. The First Impression video dataset labeled the targets in the videos using the following cutoffs: [19,24], [25,32], [33,45], [46,60], [61+]. Since we aimed to use the stratified maximum variation sampling procedure to systematically sample videos for each demographic group, we need to simplify the group categories; otherwise, we would have only very few videos within each group as you pointed out. Thus, we decided to group the videos into two larger age groups that resulted in relatively balanced numbers of videos between the two groups. This resulted in grouping the [19,24] and [25,32] groups together, and the [33,45], [46,60] and [61+] groups together, i.e., with 33 as the new cutoff.

Regarding gender and race, we directly used the labels provided by the First Impression video dataset. According to Junior et al. (2021), these labels were annotated using pretrained deep learning models and the error rates were low. Judging from the videos, “Asian” did collapse across East Asian, South Asian, and Southeast Asian targets. We agree that these are limitations of the dataset, we have now emphasized these limitations in our revised Methods. “The demographic information was provided by the First Impression video dataset, annotated using pretrained deep learning models with relatively low errors. However, there may still be slight mismatch between these computational annotations and the targets’ own self-identified age, gender, and race. The Asian category collapse across East Asian, South Asian, and Southeast Asian targets.”

Regarding video sampling, the reasons for beholding to the initial demographic proportions in the First Impression video dataset are two-fold. First, methodologically speaking, we used maximum variation sampling to systematically select videos that showed the most distinct visual, acoustic, and semantic information. This approach has been proven successful for sampling more representative stimuli and recovering more comprehensive social cognitive dimensions in our prior research (Lin, Keles, & Adolphs, 2021). Since this approach selects videos that are most distinct within the original set, if we were to sample the same number of

videos for different groups that had different numbers of videos to start with, then the selected videos will be more distinct for groups with fewer videos to start with and less distinct for groups with more videos to start with. This would introduce imbalanced social cue distributions across videos belonging to different groups. Thus, to maintain a similar level of distinctiveness across videos within different groups, we used the maximum variation sampling procedure to sample subsets of videos proportional to the initial proportions in the dataset. Second, the final demographic proportions of the selected videos also reflect the disproportional demographic groups people are typically exposed to in daily life (White 75%, Black 14%, Asian 6%), which could help generate conclusions that are more generalizable to the population.

4. One of the aspects of this work that I was most excited to dig into was the cross-cultural comparison between samples, reflecting differences in the association strength between individual inferences. As the authors write early on, "...associations between social inferences may reflect associations between personalities. For instance, people may map social reality of personalities into social perception through cultural learning, so that perceivers in different cultures that are exposed to different personality structures display different mental representations of social inferences." I will say – I think this is a big, important idea that is a *little* underdeveloped in this initial statement. I do think that the authors could go into greater detail here (potentially by expanding on the previous work they're citing) to foreground/scaffold what they're proposing and predicting. (Personally, the framing in terms of "personalities" was also a little narrow – presumably this could also be due to different cultural norms [e.g., regarding interpersonal dynamics, emotion displays, moral foundations, etc.] or linguistic differences in the meaning of specific inference concepts.)

That said, I will say that some of the data related to these cultural comparisons gave me a bit of pause – particularly with regards to their visualization in Figure 9. The framing of these network connections is given on pg. 9 – greater differences between samples reflect cases where one sample co-inferred the traits in question more so than the other sample. For example, "...when U.S. participants thought that the target person was happy, they were more likely to also thought that the person was friendly than participants in the Asian sample." This is a perfect example of what I think this analysis *could* reveal: cases where a given social inference ("happy") leads to an inference of related, but clearly distinct construct ("friendly"). And yet, many of the other examples seem considerably less meaningful – for example, "student – college," "well – spoken," "content – creator," "middle – age," etc. I doubt that it's the case that when U.S. participants infer that a target is "well" (i.e., healthy), they're also more likely to make inferences related to something "spoken" than participants in the Asian sample – or that when U.S. participants infer that a target is "content" (i.e., peaceful, tranquil, etc.), they're also more likely to make inferences related to them being a "creator" than participants in the European sample. These all seem like cases where a given multi-word phrase ("college student," "well spoken," "content creator") is more common in one sample than another... rather than a difference in the association between separate concepts.

Similarly, I'm not sure how to interpret the effects for pairs that seem like direct synonyms (i.e., "rich – wealthy," "funny – humorous"), words that are used interchangeably (even if

they ultimately have slightly different meanings; e.g., “English – British”), or words that are related in that they both describe [separate] categories of specific sociodemographic dimension (“old – middle-aged”). It would sound a bit odd to say, for example, that based on the inference that someone is rich, we might also infer that they are wealthy. The negative correlations are also a bit confusing to me: what does it mean to say that when participants in the US sample thought that a target person was White, they were less likely to also think that the person was Black than participants in the Asian sample? (Or that when participants in the US sample thought that a target person was young, they were relatively less likely to also think that the person was old than participants in the European sample?)

I don’t want to lose sight of the big picture here, but these cases did spark some questions for me about what these associations in the sparse network model are actually capturing – i.e., how many of the links in the overall network are like the hard-to-interpret or seemingly artefactual examples above?

RESPONSE: We thank the reviewer for these suggestions. First, we have now expanded our argument on how social learning may shape social inferences in the revised Introduction.

“Such high-dimensional network representations may also capture mental representations of social inferences in naturalistic contexts better than the low-dimensional latent factor representations. First, in the real world, associations between social inferences may reflect associations between personalities. This is because people may rely on social reality of personalities to inform social perception through cultural learning. For instance, exposure to different personality structures in different cultures shapes the way people conceptualize trait associations, which in turn guide social inferences^{38,39}. Thus, if the network representation better captures personality, it would also capture social inferences. More broadly, this cultural learning perspective to social inferences also predicts that other factors beyond personality, such as cultural norm and language, would lead to significant cultural variations in the network representations of social inferences^{40,41}.”

Second, regarding the cultural differences, the less meaningful examples that the reviewer pointed out, such as “student - college”, “well - spoken” etc. are not due to the separation of phrases like “college student” and “well-spoken”. For instance, “well-spoken” remained intact in our vocabulary. Instead, “well” came from multiple sources: first, lemmatization of related words such as “better”; and second, extraction from sentences where participants mentioned “well”. Therefore, although these examples are harder to interpret, they still reflect different inferences of related concepts, rather than artifact due to text preprocessing.

Third, we agree that links between synonyms are difficult to interpret as well. The results showed that most of these synonyms are more strongly linked in the US sample compared to the Asian and European samples. One plausible explanation is that participants from the U.S. have a bigger English vocabulary for describing the same concept using different words than Asian and European participants. This is likely because our U.S. participants were required to be native English speakers, but our Asian and European participants were only required to be fluent in English (since it was difficult to recruit native English speakers in these two regions). For example, the strong association between “rich” and “wealthy” in the U.S. sample indicated that the targets were frequently described with both words by different participants from the U.S. sample. The much weaker association between these two words in

the Asian sample indicated that only one of these words was used to describe the targets frequently (likely “rich” because this is a more commonly learned English word).

Fourth, the stronger negative association between “white” and “black” in the U.S. sample compared to the Asian and European samples was also meaningful. This indicated that participants within the U.S. sample more frequently mentioned the targets’ race than the participants from the Asian and European samples. Since “white” and “black” usually describe different targets, this resulted in a negative correlation between them across videos: for videos where U.S. participants wrote “white”, they were less likely to write “black”, and for videos where U.S. participants wrote “black”, they were less likely to write “white”. This finding is likely driven by the fact that race is a more salient concept in the U.S. culture, which was historically and is still being tied to systemic inequalities that are pervasive across public discourse, social media, and institutional frameworks. Consequently, racial categories may often serve as a primary concept for categorizing individuals and making social inferences for perceives in the U.S. culture.

We have now further clarified these points in the revised Results. “We compared the connections between pairwise social inferences in each of the samples in Study 2 to the U.S. sample in Study 1, using the network comparison test. We found many significant differences in the links in the networks between the U.S. and the Asian sample (Fig. 9A). For instance, U.S. participants were more likely than Asian participants to write about the target’s accent when they mentioned that the target was foreign (difference in $r = .44$, $p < .001$), and to write that the target was friendly when they mentioned that the target was happy (difference in $r = .22$, $p < .001$). We also found a significantly stronger negative association between Black and White (difference in $r = -.21$, $p = .040$) in the U.S. sample compared to the Asian sample, suggesting that U.S. participants often wrote about the race of the targets, resulting in a negative correlation between these two typically opposing racial categories from U.S. participants. This stronger negative association in the U.S. may reflect how racial categories are highly salient concepts for social perception in the U.S. We also found significant differences in the interconnections between social inferences between the U.S. and European samples (Fig. 9B). For instance, compared to European participants, U.S. participants were more likely to write about the target's accent when they mentioned that the target was British (difference in $r = .22$, $p < .001$), and to write that the target was upset when they mentioned that the target was angry (difference in $r = .18$, $p < .001$). Taken together, these findings highlight how the mental representation of social inferences may be shaped by region-specific cultural and language experiences.”

More broadly, we did observe quite many cases where a given social inference led to an inference of a related but clearly distinct construct that is more interpretable, as the reviewer expected. Previously, due to the consideration of limited space, we have only plotted 15 examples with the largest significant difference between samples, which may not present a complete picture of these results. We have now revised Figure 9 to present the results for more pairs of inferences with the most significant differences across samples. According to these results, many of the culturally different links did capture meaningful differences between samples in how one social inference led to other relevant but distinct inferences. Taken together, these findings revealed meaningful cultural differences that were typically missed by the latent factor model.

Figure 9. Edge difference between samples from different world regions. (A) Top 40 significantly different edges between the U.S. (cyan dots) and the Asian sample (orange dots). The x-axis indicates the edge weight in each sample. (B) Top 40 significantly different edges between the U.S. (cyan dots) and the European sample (purple dots).

5. Finally, I collected a few more minor main text questions that I noted while reading through the manuscript:

- On pg. 4, the authors write, “However, empirical findings comparing the two models have yet to emerge.” It’s not clear what the antecedent for “the two models” is in this paragraph – is it the latent-dimension approach and the high-dimensional network representation approach?

RESPONSE: Yes. Sorry for causing the confusion. We have now revised the main text to make this point clearer. “However, empirical findings comparing the high-dimensional network representation and the low-dimensional latent factor representation have yet to emerge.”

- On pg. 5, the authors write, “...each social inference and each relation between social inferences may need to be uniquely represented due to the distinct mechanisms underlying these inferences. For instance, different social inferences may be elicited by different perceptual inputs, stereotypes, and motivations.” While this makes sense to me, presumably it’s *also* the case that the same inference might be supported by very different (and even contradictory inputs). For instance, to use the example given by the authors on pg. 4, one might make the prediction that someone “goes to parties often” based on the (relatively disparate) inputs “likes people,” “is single and trying to meet someone,” “works as a caterer,” and “has a drinking problem” – even though those inputs are not particularly strongly related to each other. If I’m thinking about this correctly, this is a strength of the present approach that previous approaches geared towards identifying latent dimensions don’t capture.

RESPONSE: Thank you for this thoughtful observation. We agree that even the same social inference can be elicited by different inputs depending on the context. This flexible nature of social perception matches the strength of the network representation well: the network representation allows for capturing the unique and varied pathways through which an inference is formed, rather than assuming a fixed set of latent factors. We have now incorporated your inputs into our revised Introduction.

“More broadly, different social inferences in naturalistic contexts may be elicited by different perceptual inputs, stereotypes, and motivations^{26,28,42–47}. Even the same social inference may be elicited by different inputs depending on the context (e.g., “likes people” may be inferred from “goes to parties often” or “works as a caterer”). Furthermore, different social inferences may also elicit one another through heuristics (perceived attractiveness elicits perception of other positive characteristics such as intelligence^{48–51}) and other conceptual biases (stereotypical relations between feminine-looking individuals and characteristics traditionally associated with females^{46,52}). Thus, the complex relations between social inferences in naturalistic contexts may be better captured by high-dimensional network representations, which allow for unique variations across inferences, than the latent factor representation, which assumes a large amount of common variation across inferences.”

- On pg. 8, the authors write, “We first quantified multi-modal features of the targets in each video, including the visual features of facial identity.” What does “visual features of facial identity” refer to? I’m assuming it’s sociodemographic features like race, gender, and age. If so, how accurate and reliable is this tool?

RESPONSE: Visual features of facial identity refer to structural features of the face. These are 512 uninterpretable features quantified by a facial recognition pre-trained neural network. These 512 features capture the structural features of a face so that they can distinguish photos of one person (in different lighting, angles, with different makeups) from those of another person. We have now replaced the term of “facial identity” with “facial structure” for clarity.

- Also on pg. 8, the authors state that they “L-2 normalized the features before sampling.” I wasn’t immediately familiar with the concept of L-2 normalization; what does it refer to?

RESPONSE: L-2 normalization is a technique that scales the feature vectors (e.g., the 512 facial structural features we mentioned above) so that the total length of the vector sums up to 1. Specifically, we divided each component of a vector (e.g., each value in the 512-dimensional vector) by the square root of the sum of all squared components. This normalization helps ensure all features (e.g., facial structural features, action unit features, auditory frequency features, etc.) are on a comparable scale and prevent the features that are represented with a greater number of components from dominating the analysis.

- On pg. 11, the authors say that they “excluded a participant if they did not meet our inclusion criteria, failed one or more attention checks, or having more than half of the videos excluded because all words across all pauses in those videos were meaningless.” This probably isn’t a big deal, but I just wanted to note that I stumbled on this sentence a few times, because the concept of “pauses” in the procedures had not yet been explained. (Similarly, at this point, the reader doesn’t know what the attention checks are or what a meaningless word would be.)

RESPONSE: Sorry for causing the confusion. We have modified the corresponding section to make it clearer. “ii) we excluded participants who did not meet our inclusion criteria, failed one or more attention checks (see Procedures below), or had more than half of their videos excluded due to providing only meaningless words as responses”.

6. A few additional questions regarding the figures:

- What do the color-coded pentagons in Figure 1B represent?

RESPONSE: The color-coded pentagons in Figure 1B indicate the communities. Since we have not mentioned anything about communities yet in the Introduction, we have removed these color-coded pentagons in Figure 1B to minimize confusion.

- What do the inferences “bos” and “ci” in Figure 3 refer to?

RESPONSE: Thank you for pointing this out. The inferences “bos” and “ci” in Figure 3 were artifacts caused by the automatic stemming algorithm (i.e., converting a word to its base form). Specifically, “bos” represents “boss,” and “ci” represents “cis” as in “cisgender.” We have now corrected these words in Figure 3.

Figure 3. Factor loadings of exploratory factor analysis in Study 1. Exploratory factor analysis on the co-occurrence of the 2,926 social inferences (rows, only the top 5 inferences per dimension are plotted here) freely generated by participants based on 444 naturalistic videos indicated 25 dimensions (columns) optimally underlie the data. Positive loadings are annotated in red; negative loadings in blue. The darker the color is, the greater the absolute value of the factor loading is.

- Figure 6 (and the accompanying text on pg. 24) was a little hard for me to grasp. I *think* I'm correctly understanding now that the rows in Figure 6 represent the top five inferences at each time point in terms of change in strength centrality from the previous timepoint. (Therefore, Time1 would just be the five inferences with the highest overall strength centrality?) So the top five at Time1 are "pretty," "bodybuilder," "muscle," "gym," and "trainer;" the top five at Time2 are "man," "woman," "male," "female," and "pretty" again, etc. I think I was tripping on the "total of 19 unique social inferences" aspects of this figure – i.e., it could have been 10 or 20 inferences per time point, but the authors went with 5 per time point for presentation's sake. Is that accurate?

If so, when the authors say, "We found that inferences made at the earlier timepoints were more about physical appearances (e.g., muscular), then about social categories (e.g., gender), followed by descriptions of what the person was talking about (e.g., advise), and finally inferences of abstract personalities (e.g., empower)," these qualitative characterizations are ultimately subjective judgments being made by the authors, right? The trajectory they're describing makes narrative sense, but it seems like it would be just as logical to say that participants first judged whether targets were fitness enthusiasts, then judged their gender identity, then judged their rhetorical skill, and finally judged their power/dominance. I don't mean to be silly; I think my point is just that these interpretations feel a bit overstated, particularly given how these data are presented.

RESPONSE: Sorry for the confusion. Your understanding is correct. We have revised Figure 6 legend to clarify this figure: "Inferences in the rows are sorted from bottom to top in descending order according to the increase in centrality at each time point." For each time point, a few hundred unique inferences were made. We selected the top 5 non-overlapping words for each pause for demonstration.

The reviewer is correct that our interpretation of the results is subjective. We have now revised the Results to more objectively described the patterns observed in the figure. "The network representation not only revealed the dynamics of social perception in terms of what descriptions people wrote together regarding a target but also what descriptions people wrote over time. We estimated an SNM for inferences entered at four different time points: the first word participants wrote based on the first frame of the video, additional words they wrote based on the first frame, the words they wrote after viewing half of the video, and the words they wrote after viewing the entire video. For each network, we computed the strength centrality of each inference, which is the sum of all edges connecting that inference to other inferences. Using these strength centralities, we assessed the change in connections between social inference across the four timepoints (Fig. 6). We found a greater increase in strength centrality of physical appearances/fitness (e.g., muscular) and gender identity (e.g., female) from timepoint 1 to timepoint 2, of speech content/rhetorical skill (e.g., mentor) from timepoint 2 to timepoint 3, and of abstract personalities/power (e.g., executive) from

timepoint 3 to timepoint 4. These findings suggest that the social inferences that were central to generating other new inferences changed over time, shifting from more physical inferences to more abstract inferences. Moreover, physical inferences continued to show high strength centrality across all four time points, suggesting their sustained influence on the descriptions perceivers wrote across all time points.”

- I don't see a distribution corresponding to the Theory II model of the European dataset in Figure 7. Is it missing? Is it overlapping with the Theory II Asian distribution? If so, is there a way to present this that's a bit clearer, visually?

RESPONSE: The distributions for the SNM from the two samples are indeed highly overlapped. We have now changed the color scheme to highlight the difference between the two samples.

Figure 7. Prediction accuracy of correlations between inferences from two models. The x-axis indicates the error, i.e., standardized root mean squared residual (SRMR), between the predicted correlations among social inferences based on a specific model (LFM: latent factor model, in green; SNM: sparse network model, in purple) and the actual observed correlations among social inferences in each of the samples.

- The caption for Figure 9 indicates that the figure presents the “Top 10 significantly different edges between the U.S. (cyan dots) and the Asian sample (orange dots),” as well as the top 10 different edges between the U.S. and European samples. However, there 15 rows in each panel of the figure.

RESPONSE: Thank you for pointing out this typo. We have now added more examples to Figure 9 in response to your comment #4 above, providing results for the top 40 different edges for each comparison.

7. Finally, I noted a few typos:

- On pg. 4, in "...the network representation explains individual difference in terms of the various causal links between different behavior and cognition," should the last phrase be "different behaviors and cognitions"? (Also, replacing "different" with "specific" might clarify the meaning here.)

- On pg. 7, the authors write, "...we targeted the First Impression video dataset that were gathered by a prior research." I think "by a prior research" should be "in prior research."

- On pg. 11, the authors write, "Since it would take too long for each participant to evaluate each of the 444 videos. We randomly assigned these videos into modules, each with 12 videos." These two sentences should be joined together in one.

- On pg. 31, the authors write, "...when provided with a rich set of information that naturally co-occur, a recent research showed that perceivers use distinct sets of information to form impression of different traits." I'd suggest rephrasing "a rich set of information that naturally co-occur" as "a rich set of naturally co-occurring information," and further, changing "a recent research" to just "recent research."

RESPONSE: Thank you for pointing out these typos and offering helpful suggestions. We have incorporated these into our revised manuscript.

"...the network representation explains individual differences through distinct causal links among behaviors."

"...we targeted the First Impression video dataset that were gathered in prior research"

"Since it would take too long for each participant to evaluate each of the 444 videos, we randomly assigned these videos into 37 modules, each with 12 videos."

"However, when provided with a rich set of naturally co-occurring information, perceivers use distinct sets of information to make different social inferences⁴²."

Reviewer 3

Reviewer #3 (Remarks to the Author):

In this manuscript, the authors ask whether social inferences are well described by a few latent variables or require more complex models. The authors find that social inferences in response to video clips are poorly explained by latent dimensions, and continue to advance a network model which describes the data better.

In the present studies, participants watched clips and responded freely over multiple timepoints of the clips. The authors also investigated the temporal dynamics of social inferences, and examined cross-cultural differences in network dynamics by sampling US American, Asian, and European participants.

This paper uses an interesting methodological approach to tackle an intriguing problem. I would like to highlight positively the use of open-science methods, the use of naturalistic, complex stimuli, repeated measures, and a network approach. Although I find this paper interesting, I am

not sure about the conclusions that can be drawn from the present approach. See my detailed points below.

Major

1. I am not an expert in this particular technique, but 'excluding words that only one participant mentioned and mentioned in less than 1% of the videos', reduces your data from more than 9000 unique words to less than 3000. That seems like an usually large proportion of data, and I also fail to find this in the preregistration (please point me to the relevant section, if it can indeed be found there).

RESPONSE: The reviewer's concern is well taken. This exclusion criterion was not preregistered but it is a common cutoff used for natural language processing to reduce noise and enhance interpretability. Excluding words mentioned by only one participant or in less than 1% of videos helps to minimize idiosyncratic responses that could just be outliers and disproportionately affect results. For instance, in another well-cited work that uses natural language processing to analyze spontaneous attributes participants generated for different social groups (Nicolas, Bai, & Fiske, 2022), the authors even excluded all words that were provided by less than 5 participants, reducing their original dataset size from 14,400 descriptions to 10,656 descriptions (an exclusion rate of 26%). In our case, our exclusion procedure reduced our original dataset size from 225,184 descriptions to over 210,000 descriptions (an exclusion rate of less than 7%). We have now added a reference to justify this exclusion criteria in our revised Results: "The 1% exclusion criterion was based on common thresholds in the literature of natural language processing⁷³".

2. Was it actually 'social inference' what people did or just descriptions of these videos? I failed to find the specific instructions used in the prompts, and I wonder whether all these words can be classified as social inference or whether parts of them constituted unrelated comments.

RESPONSE: Thank you for bringing up this concern. Participants were instructed to describe their impressions of the people in the videos, not the videos themselves, to ensure the focus remained on social inference. Specifically, the instruction was, "Please write down ALL words that come to your mind about the person". Nevertheless, we agree with the reviewer that even with this instruction, some participants may still write words that are not about the person but just descriptions of the videos. Applying the exclusion criteria to remove words mentioned by only one participant or in less than 1% of the videos should help remove these outliers. The remaining words are largely consistent with our instructions about social inferences, as the examples shown in Figure 3, Figure 5, Figure 6, Figure 8, and Figure 9.

3. Since participants did not structure—by definition—their inferences, it's hard to know which words they thought were central and which other words were peripheral. I suppose that a for instance 2-dimensional model would still posit that peripheral descriptions will revolve around their two dimensions, but nonetheless: Since the authors explicitly gave the option to fill in 10 textboxes, participants may understand that as many as 10 words may be important to the description.

RESPONSE: The reviewer rightly pointed out that participants were not instructed to structure their responses in terms of how central or peripheral they each thought the

descriptions were about the targets. This limitation also applies to previous studies that demonstrated the low-dimensional latent factors, where participants were asked to rate targets on a list of attributes without ranking the attributes in terms of their centrality to the descriptions of each target. The low-dimensional latent factor models do imply the centrality of different inferences, but not in terms of their centrality to the descriptions of the targets but their centrality (i.e., correlations) in relation to the latent factors. Although in our experiment we provided 10 textboxes to participants, participants typically wrote fewer than 3 words. This alleviates the concern that participants might infer that as many as 10 words were important for the description.

4. You report % variance explained numbers for the 25 and the 100 dimension model, yet I do not find a comparable number based on your network model. I understand that the error of the SNM is lower, but how can I evaluate this quantitatively?

RESPONSE: The reviewer raised an excellent question. The percentage of explained variance in exploratory factor analysis (e.g., the 15% variance explained by the 25 latent factors in the U.S. data) is a unique metric specific to exploratory factor analysis (EFA). EFA assumes that each item (i.e., each social inference in our case) is driven by two types of variances, one that is common across all the items, and the other that is unique to each item. The explained variance typically reported in EFA refers to only the common variance. That is, EFA only cares about how much the latent factors explain the way all items increase or decrease together (i.e., common variance). Thus, the 15% variance explained by the 25 latent factors reported here in fact referred to the 15% common variance.

However, the network model does not assume that each item is driven by both common and unique variances. Thus, the percentage of common explained variance is irrelevant to network models. That's exactly why we needed to find a model fit index that is relevant to both models to directly compare them. The standardized root mean square residual (SRMR, see Figure 7) metric is well-suited for this purpose. The SRMR measures how many errors each model made when we used the model outputs (e.g., the factor loadings from the latent-factor model, and the partial correlations from the sparse network model) to re-estimate the full covariance matrix between all social inferences in our dataset. Using this metric, we can directly compare which model provides a more valid parsimonious representation of our observed data.

5. Although the idea of studying temporal dynamics in your network model is interesting, it seems rather obvious that surface-level characteristics are described first and content-related characteristics described later. It is good that your model captures this intuition, but I wonder about the contribution of these findings.

RESPONSE: The reviewer's concern is well taken. We would like to clarify that the essence of the strength centrality analysis is not to reveal which type of inferences were written earlier than others but to identify the influence of one inference on other inferences in each given time point. An inference that is more densely connected with other inferences in the network at a given time point suggests that this inference may be more important for eliciting other inferences that participants wrote down within the same time point. We have now revised the Results to clarify the interpretation and implication of these strength centralities.

“For each network, we computed the strength centrality of each inference, which is the sum of all edges connecting that inference to other inferences. Using these strength centralities, we assessed the change in connections between social inference across the four timepoints (Fig. 6). We found a greater increase in strength centrality of physical appearances/fitness (e.g., muscular) and gender identity (e.g., female) from timepoint 1 to timepoint 2, of speech content/rhetorical skill (e.g., mentor) from timepoint 2 to timepoint 3, and of abstract personalities/power (e.g., executive) from timepoint 3 to timepoint 4. These findings suggest that the social inferences that were central to generating other new inferences changed over time, shifting from more physical inferences to more abstract inferences. Moreover, physical inferences continued to show high strength centrality across all four time points, suggesting their sustained influence on the descriptions perceivers wrote across all time points.”

6. This may very well reflect my own lack of understanding, but: Could it be that your results are due to autocorrelation between people's inputs? For instance, if participants' inputs are quite consistent but not explained by any underlying factor, a model which suggests low variation in responses will fare well.

RESPONSE: The reviewer raised an interesting possibility. If high autocorrelation refers to a participant's responses being correlated (e.g., writing synonyms), this will instead favor the discovery of latent factors. This is because the low explained variance in EFA does not refer to low variation in the responses but that there is a small amount of variance that is common across the responses that the latent factors can explain. If all responses were synonyms or that there are subgroups of inferences which are synonyms, this would instead boost the common variance across responses and resulting in a better fit of the latent-factor model.

7. Please read the manuscript carefully for grammar and typos. E.g. line 171: 'a prior research'. Furthermore, line 242 has a fullstop where it should have a comma, line 277 should say 'groups' instead of 'group', 'experimental paradigm' instead of 'experiment paradigm' on line 701, 'thought' in line 551 should be 'think' (this is a recurring mistake throughout the manuscript), and many more issues. Also, be careful with using 'Figs.' instead of 'Fig.', for instance in line 684.

RESPONSE: Thanks for pointing out these grammatic errors. We have now corrected them throughout the manuscript.

Minor

1. Please change the R command in line 168 to a verbal description of the procedure.

RESPONSE: We have indeed provided a verbal description of the procedure in the sentence preceding the R command. To avoid confusion, we have now removed the R command.

2. although your figures are visually appealing, they mostly are of low resolution/seem washed out. Could you please export them in higher resolution?

RESPONSE: Thank you for pointing this out. We have checked our local file and found that the low image resolution might be caused by the server of the journal submission system. We will contact the journal to figure this out.

REVIEWER EXPERTISE:

Reviewer #1: Computational modeling, person perception

Reviewer #2: Person/social perception

Reviewer #3: Computational modeling, person perception

REVIEWER REPORTS:

Reviewer #1 (Remarks to the Author):

I appreciate the authors detailed responses to my comments, but I don't think they adequately engaged with some of the broader points. Below, I provide further detail.

1. In their response to Comment #5, the authors miss an opportunity to highlight the broader significance of their findings for the study of impression formation. Instead, they raised a different question about causal mechanisms: "What the relationships between personality dimensions and social cognitive dimensions are is indeed a very interesting question that remains to be addressed. A related and even deeper question is how we should conceptualize these psychological dimensions: are they fixed internal forces or emergent properties?"

I am not sure what the authors are distinguishing here with personality and social cognitive dimensions. If they are referring to warmth and competence, I would say that these too are personality traits but ones that did not emerge as clearly in the present work as the big five. Additionally, while the nature of personality traits—whether biologically determined or emergent—is an ongoing debate, it does not seem central to interpreting the present findings, which suggest the Big Five are central to social inferences. Importantly, the present evidence of the emergence of the Big Five in spontaneous trait inferences was anticipated by prior theoretical work titled: The five-factor model describes the structure of social perceptions (Srivastava, 2010). It also supports decades of research and focus on Big Five impressions in the field of interpersonal perception.

Considering that perceptions made of these more natural dynamic stimuli more closely resemble those made of people rather than static stimuli, the findings also raise an important question about the validity of inferring real-world effects from the study of perceptions of static stimuli.

RESPONSE: We thank the reviewer for raising this important point and for the relevant references. We agree with the reviewer that we should highlight the broader significance of our findings in terms of the central role of Big Five dimensions in social inferences. We have now highlighted this in our revised Discussion (please see detail attached below). We also agree with the reviewer that our findings highlight the importance of using more dynamic and naturalistic stimuli to better understand real-world effects.

We would like to clarify why we were trying to explain the distinction between personality dimensions and social cognitive dimensions. We made that distinction with the intend to address the question raised in the reviewer's Comment #5, "The authors came to a different conclusion from the data, so I think it is important for them to address why these communities look like but are not the big five." We attempted to address why the seven communities of social inferences in

our finding (Fig. 5) may not perfectly align with the Big Five personality traits, even if they partially overlap as previously shown (Srivastava, 2010). We think this is because the two sets of dimensions intend to explain different phenomena. Conceptually speaking, personality dimensions, such as the Big Five personality dimensions, intend to explain the true individual differences in people's behavior (Goldberg, 1993). Even though, practically, they are often measured and derived based on people's perception, either the individual's self-perception, or, as the reviewer pointed out previously, other people's perception, "The big five comes from words people use to describe each other (e.g., Saucier & Goldberg, 1996)"; conceptually, these dimensions intend to explain causes of true individual differences (McCrae & Costa, 2008). On the other hand, social cognitive dimensions are often interpreted as summarizing people's perceptions of others rather than causes of others' behavior. These perceptions may not reflect others' true personality. Thus, we were hoping that clarifying this distinction helped address why we refrain from making stronger claims about the relationships between the seven communities in Fig. 5 and the Big Five personality dimensions.

Despite this conceptual distinction, we fully agree with the reviewer that, practically, communities of social inferences (social cognitive dimensions) and the Big Five personality dimensions are likely highly similar. We think at least two mechanisms may support this similarity.

First, the lexical foundation of both domains likely contributes to this alignment. Both the Big Five model and models of social cognitive dimensions are based on analyses of natural language descriptors of human behavior (Goldberg, 1993; John & Srivastava, 1999). This lexical overlap means that both self-perceptions and perceptions of others are filtered through the same fundamental vocabulary of trait descriptors.

Second, there may be a deeper psychological basis for this similarity: interpersonal perception and self-perception are likely interrelated. For example, perception of others may be shaped by one's self-concept (e.g., self-reported personality traits), which in turn is also shaped by how others view us. Supporting this, prior research has shown that perceiver effects—heterogeneity in how individuals perceive others—are described by a five-factor structure (Srivastava et al., 2010).

We have now highlighted the role of the Big Five in social inferences in our revised Discussion:

"Second, the communities identified based on social inferences (Fig. 5) also bore resemblance to the Big Five personality dimensions. For instance, the orange community (containing inferences such as nerdy and geeky) and the red community (containing inferences such as musician and strange) were similar to the two facets of openness – intellect and aesthetic sensitivity; the purple community (containing inferences such as friendly and happy) was similar to agreeableness; and the green community (containing inferences such as depressed and lonely) was similar to neuroticism. This similarity between social inference dimensions and personality dimensions was consistent with prior work showing that perceiver effects in social inferences are structured by the Big Five dimensions^{92,93}. This similarity may be explained by the shared lexical foundation of studies on social inferences and those on personality dimensions, which both analyzed verbal descriptors of human behavior⁹⁴. It may also be driven by the bidirectional relationship between self-perception and perception of others: how we perceive others is shaped by our self-perception, and vice versa^{95,96}."

2. The authors did not address the issue I raised in comment #4. My concern about the background information being an alternative explanation for the additional domains of social perception, compared to static stimuli remains.

RESPONSE: Thank you for pointing out the ambiguity in our previous response. We appreciate the opportunity to clarify our perspective. Our study focuses on investigating the content and organization of social inferences. Our data could not precisely pinpoint *WHY* we observed these different contents/organizations than prior work, and that would be beyond the scope of the current research. However, we agree it is helpful to discuss potential mechanisms.

We agree with the reviewer that the additional background information in our research beyond prior work may have contributed to the additional domains found here. In our prior response, we mentioned a recent study by Connor et al. (2024), which used static images with diverse backgrounds, found way more dimensions of social inferences than prior work using images without backgrounds. This study suggests that indeed some of the additional dimensions we found here may also be driven by the diverse backgrounds in our stimuli.

We also think that other factors such as the dynamics of the stimuli may also have contributed to the additional domains found here. For example, we identified latent factors such as dominance, which did not emerge in the findings of Connor et al. (2024) even with diverse backgrounds. The top descriptors for this latent factor in our study included terms such as *express*— and inference likely driven by dynamic auditory and behavioral cues rather than the background.

We have now highlighted both potential mechanisms in our revised Discussion: *“For instance, compared to studies using artificial images, our paradigm revealed additional inferences that were likely driven by the contextual backgrounds (e.g., the inference of “rural”) and dynamic information (e.g., the inference of “express”). This highlights the importance of modalities beyond visual cues, such as auditory and semantic information, in shaping social inferences.”*

3. Finally, I urge the authors to double check that all citations support claims.
 - a. the opening sentence states that people make a wide variety of social inferences in daily life but the citations to support this are papers on social inferences from faces (1-4)—not of people in daily life.
 - b. Citation 87 and 92 do not appear to support the statement: “Most prior work conceptualizes latent factors as fixed internal forces that cause psychological processes, such as the Big Five personality traits causing individual differences in behavior^{87,92}.”
 - c. Citation 93 does not support a 3 dimensional model—it is a paper validating the Big Five—“The results are straightforward, showing convergent and discriminant cross-observer and cross instrument validation for all five factors.” (McCrae & Costa, 1987, p. 86)
 - d. The other citation for a three-factor model of social inferences (19) comes from the study of stereotype content of social group labels, not inferences about individuals.

RESPONSE: Thank you for these suggestions.

Re (a), we have now included the reference from Fiske & Cox (1979) in the revised manuscript to show the diverse inferences people make of others in daily life.

Re (b), we have now cited Nettle (2011) and Oosterhof & Todorov (2008), which argue for functional/evolutionary bases of these dimensions: “*Most prior work conceptualizes latent factors as fixed internal forces that cause psychological processes, such as personality dimensions⁸⁴ and social cognitive dimensions¹⁰.*” We would also like to clarify that conceptualizing factors as fixed internal causes is inherent to the logic of factor analysis (Borsboom et al., 2004).

Re (c), if we understand correctly, we think the reviewer is referring to the citation in this sentence, “*Most prior work conceptualizes latent factors as fixed internal forces that cause psychological processes, such as the Big Five personality traits causing individual differences in behavior^{89,93}.*” We may have misunderstood which sentence the reviewer was referring to but in this sentence, we were indeed using citation 93 to support the Big Five.

Re (d), if we understand correctly, we think the reviewer is referring to the citation in this sentence, “*However, prior work on the latent factors underlying social inferences discovered ever changing sets of dimensions, from the Big Two^{12,91} and Big Three^{19,94}, to fourteen and even forty dimensions^{24,25}.*” This sentence refers to dimensions of social inferences, and inferences of social groups fall within the domain of social inferences, so we think citation 19 is appropriate.

Reviewer #2 (Remarks to the Author):

The authors' revision was very responsive to the questions and critiques raised in the first round of review. In particular, I found their response letter to be particularly comprehensive and informative. From my perspective, the manuscript has been usefully clarified and improved. I have a few remaining stray thoughts/questions that I've listed below, which I hope the authors will consider as they move forward. Thank you once again for the opportunity to consider this work!

1. In their response to my comments, the authors posited a plausible account of why some of the cross-cultural differences may have emerged in Study 2. As they write, “...participants from the U.S. have a bigger English vocabulary for describing the same concept using different words than Asian and European participants. This is likely because our U.S. participants were required to be native English speakers, but our Asian and European participants were only required to be fluent in English.” I don't know that I'd fully processed how that difference between samples might impact the observed network representations. It's not a major issue, but if the authors do indeed find this to be a reasonable explanation, it's something worth drawing the reader's attention to more explicitly. (More generally, I hadn't considered how these differences in correlations could reflect both a) an increased association between concepts in one sample vs. another and b) increased discussion overall of a given topic in one sample vs. another, so the authors' explanation was helpful.)

That said, I wanted to try to estimate the magnitude of the difference the authors are describing here in terms of vocabulary/usage differences across samples. The authors state that after processing/exclusion, there were 2964 unique words used to describe the Study 1 videos, which,

with 1598 participants comes out to 1.85 unique words per participant. If what the authors are proposing is accurate, the number of unique words per participant should be lower in both the Study 2 Asian and European samples. By this metric, it is in the European sample ($1319/792 = 1.66$ unique words per participant), but not in the Asian sample ($1351/651 = 2.07$ unique words per participant). That said, maybe I'm not thinking about this the right way and/or this isn't the best metric of vocabulary (or if differences of this magnitude on this metric are meaningful). In any case, it would be helpful to get the authors' take on this.

RESPONSES: We thank the reviewer for this question and the insightful analysis. We would like to clarify the network analysis results and our proposed plausible account.

The sparse network analysis results do not reflect the raw correlation (zero-order correlation) between two words based on their co-occurrence across different videos. Instead, the links in sparse networks reflect partial correlations – the correlation between two words that cannot be accounted for through other words. We have now revised the manuscript to explain this (please see relevant revision attached below). Thus, the difference in the links between two given words in two different sparse networks from two cultures does not reflect their zero-order correlation differences, but their partial correlation differences. In fact, the zero-order correlations between words in different samples were highly similar. However, the sparse network models capturing the partial correlations showed interesting differences between cultures. A stronger partial correlation in the US sample compared to the European and Asian samples suggests that the correlations between words in the US sample are less explainable through other words than the European samples and Asian samples. This may be a result of US participants, as native English speakers, writing words that were referring to more distinct concepts due to their larger English vocabulary. That is, the more distinct concepts mentioned, the stronger partial correlations the sparse network is likely to preserve. We apologize that our previous proposed account has not accurately express this –it is not bigger English vocabulary for describing the same concepts but bigger vocabulary for describing distinct concepts. We have now corrected this in our revised Results:

“We compared the connections between pairwise social inferences in each of the samples in Study 2 to the U.S. sample in Study 1, using the network comparison test. We found many significant differences in the links of the networks between the U.S. and the Asian sample (Fig. 9A), indicating regional variations in the partial correlations among trait inferences (i.e., correlations between two inferences that were not explainable through other inferences). For instance, the unique correlations between accent and foreign (difference in $r = .44$, $p < .001$) and between friendly and happy (difference in $r = .22$, $p < .001$) were both significantly stronger in the U.S. than Asian participants after accounting for other inferences. We also found a significantly stronger partial correlation between Black and White (difference in $r = -.21$, $p = .040$) in the U.S. sample compared to the Asian sample, suggesting that race was more salient in U.S. participants' descriptions, and that the (negative) conceptual association between different racial categories was less explainable through other inferences. We also found significant differences in the partial correlations between social inferences in the U.S. and European samples (Fig. 9B). For instance, after accounting for other inferences, U.S. participants showed a stronger unique link between mentioning a target's British origin and describing their accent (difference in $r = .22$, $p < .001$), as well as between describing targets as angry and upset (difference in $r = .18$, $p < .001$). Taken together, these findings highlight how

the mental representation of social inferences may be shaped by region-specific cultural experiences.”

We also emphasized this in the Discussion:

While the low-dimensional latent construct approach often reveals universal structures underlying social inferences across cultures and developmental conditions^{59,76,78–82}, the network representation revealed significant differences in the unique associations (partial correlations) between social inferences across world regions (Fig. 9).

However, the number of concepts mentioned may not be directly reflected by the number of unique words written. For instance, two participants could write down the same number of words but one could be using many different words to describe the same concept (e.g., synonyms, antonyms, or conceptually related words) and the other could write about a lot of distinct concepts (conceptually less related words) while using only a few words to refer to each concept. Therefore, the calculation of the number of unique words per participant is informative but their relation with the number of distinct concepts mentioned is less clear. Instead, we tried leveraging natural language models to analyze the conceptual relation between words. In brief, natural language models convert a specific word into a vector of numbers, and that this vector reflects the semantic meaning of the word. Words with more similar semantic association (conceptual association) have a higher cosine similarity. Consistently with our speculation, the average similarity between unique words written by European participants (0.269, SD = 0.025) and Asian participants (0.270, SD = 0.025) was greater than those by the US participants (0.259, SD = 0.024). The differences indicate medium-to-large effects (Cohen’s $d = 0.45$ between US and Asian samples; $d = 0.41$ between US and European samples) and thus shows that the semantic concepts reflected in the words written by US participants may be more diverse than those by European and Asian participants, leading to more partial correlations between words that cannot be explained through other words. We have now clarified this plausible account in the revised manuscript:

“The overall greater number of unique associations in the US network may reflect more diverse concepts in participants’ descriptions compared to Asian and European participants. Word embedding analysis revealed lower average cosine similarity between inferences from the US participants ($M = 0.259$, $SD = 0.024$) compared to Asian ($M = 0.270$, $SD = 0.025$) and European ($M = 0.269$, $SD = 0.025$) participants, showing a medium-to-large effect size (Cohen’s $d = 0.45$ between US and Asian samples; $d = 0.41$ between US and European samples). These results suggest that US participants may draw inferences from a broader conceptual space. This lexical diversity may thus produce more irreducible partial correlations between inferences. Taken together, these findings highlight how the mental representation of social inferences may be shaped by region-specific cultural experiences.”

2. On the subject of the descriptions/inferences generated spontaneously by participants, it strikes me that (as in other related work like Connor et al., 2024), the descriptions given are on the whole, somewhat positively skewed (i.e., I’m not seeing a lot of “untrustworthy,” “ugly,” “dumb,” “weak,” etc.). Beyond that, while participants are making some inferences

regarding character traits (i.e., things like intelligent and kind appear among the inferences in Figure 3), there are far more inferences about stable aspects of identity (i.e., age, gender, etc.), social roles (i.e., farmer, businessman), and emotionality (i.e., upset, angry, happy). Is there any way of quantifying the valence skew (either overall or within the most trait-like words) in the set of inferences? Moreover, if the set is positively skewed, is there any concern that this reflects some degree of social desirability bias? (Perhaps participants are inferring certain features that they don't feel it's appropriate to report on.)

RESPONSE: The reviewer raised a very interesting point. Indeed, a lot of prior research showed evidence of positivity bias. To quantify the valence of words collected in our study, we used natural language processing as mentioned in our response to your comment #1. Specifically, we performed a sentiment analysis, in which a sentiment score was computed for each word written by participants. The sentiment score ranges from -1 (negative) to 1 (positive). Based on the analysis of the 23,000+ social inferences in our data, we found that the positivity bias was weak in our data, with a mean sentiment score very close to zero in all three samples (US: 0.08, Asian: 0.08, Europe: 0.09). The distribution shows that most words were neutral and that there was a wide spread in the valence of the inferences. This different pattern revealed from our data than prior work (e.g, Connor et al., 2024) may be explained by the more diverse information streams participants had access to here (e.g., the diverse environments the target individuals were situated in, the diverse demographic backgrounds and topics discussed).

We have mentioned these in the revised Discussion:

“Additionally, unlike prior studies⁸⁴, we did not observe a significant social desirability bias, suggesting more diverse trait inferences in naturalistic settings. Sentiment analysis, which assigns each word a score ranging from -1 (negative) to 1 (positive), revealed a mean valence close to zero (US mean = 0.08, Asian mean = 0.08, European mean = 0.09), with a wide spread of valence score and most words being neutral.”

A. US Sample

B. Asian Sample

C. European Sample

- When reading Point #4 from Reviewer 1 (and the authors' response), I got to thinking about the other social cues that are present in the videos (i.e., cues to social class, body shape and size, environmental context, etc.) that were not considered in the feature quantification / extraction step described on pages 9 and 10. (The Vikings shirt that Bean Woman in Fig. 2A is wearing is an important cue for me, personally!) Just thinking this through... what's the consequence of not characterizing something like social class and accounting for it in the

maximum variation sampling? Is it just that there might be less variation on that feature across the full stimulus set than there is for the features described on page 9?

RESPONSE: We acknowledge that there are additional sources of social information that were not included in our maximum variation sampling and may also play important roles in impression formation. The purpose of the maximum variation sampling procedure is to increase the generalizability of our results to the stimulus space – the more diverse stimuli (more diverse features) we sample, the more diverse situations within those feature spaces our results can generalize to. Therefore, not characterizing some potentially important features in the maximum variation sampling procedure may limit the generalizability of our results along those features (e.g., social class). However, we would like to highlight that there are too many potential features that may be relevant to impression formation, and it is practically challenging to quantify all of those in naturalistic videos. Therefore, we prioritize the three types of features across three modalities that are unique to naturalistic, dynamic videos, and that can be relatively objectively and consistently quantified (visual features: facial structure, facial expressions, head pose; auditory features: frequency, spectrum, amplitude, and temporal features; semantic features).

4. Finally, I just had a small wording note. On pg. 36, the authors write, “The network representation also highlights the importance of centrality in understanding social inferences. That is, the importance of a social inference in terms of its connections to other social inferences.” The second sentence here is a fragment – can these two thoughts be joined?

RESPONSE: Thank you for raising this point, we have modified the sentence to make it clearer. *“The network approach demonstrates how a social inference’s psychological significance can be understood through its centrality—the extent to which it connects to and influences other judgments within the impression formation process.”*

Reviewer #3 (Remarks to the Author):

The authors have addressed my questions thoughtfully and to my satisfaction.

RESPONSE: Thank you for your efforts and insightful suggestions.

REVIEWERS' COMMENTS:

Reviewer #1 (Remarks to the Author):

I appreciate the author's responsiveness and engagement. In the current manuscript, the authors have adequately addressed my previous comments and made important connections with the broader literature. My thanks to you and the authors for asking me to review this manuscript.

Reviewer #2 (Remarks to the Author):

The authors have addressed all of my remaining questions and comments. Thank you again for your careful and comprehensive revision of this manuscript -- and for the opportunity to read this work!

RESPONSE: We sincerely thank both reviewers for their constructive feedback and kind acknowledgment of our revisions.